# Off-shell strings II: Black hole entropy

**Amr Ahmadain⋆ and Aron C. Wall†**

Department of Applied Mathematics and Theoretical Physics (DAMTP),
University of Cambridge, Cambridge, United Kingdom

⋆ amrahmadain@gmail.com , † aroncwall@gmail.com

## Abstract

In 1994, Susskind and Uglum argued that it is possible to derive the Bekenstein-Hawking entropy $A/4G_N$ from string theory. In this article we explain the conceptual underpinnings of this argument, while elucidating its relationship to induced gravity and ER=EPR. Following an off-shell calculation by Tseytlin, we explicitly derive the classical closed string effective action from sphere diagrams at leading order in $\alpha'$. We then show how to use this to obtain black hole entropy from the RG flow of the NLSM on conical manifolds. (We also briefly discuss the more problematic "open string picture" of Susskind and Uglum, in which strings end on the horizon.) We then compare these off-shell results with the rival "orbifold replica trick" using the on-shell $\mathbb{C}/Z_N$ background, which does not account for the leading order Bekenstein-Hawking entropy—unless perhaps tachyons are allowed to condense on the orbifold. Possible connections to the ER=EPR conjecture are explored. Finally, we discuss prospects for various extensions, including prospects for deriving holographic entanglement entropy in the bulk of AdS.

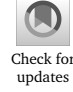

# 1  Introduction

As shown by Bekenstein and Hawking [1,2], the black hole entropy in general relativity is proportional to the area:

$$S = A/4G_N \,, \tag{1}$$

in units where $\hbar = c = k = 1$. There are many famous derivations of (1) in string theory, the most well-known of which is the one by Strominger and Vafa where explicit counting of the microstates was done in terms of the BPS states of a supersymmetric black hole [3]. Among them is a notorious 1994 article by Susskind and Uglum [4] (henceforth S&U), which claims to derive (1) from the string theory worldsheet perspective, for black holes which are far from extremal (in fact they take the infinite mass limit so as to calculate in Rindler spacetime). While the S&U paper has over 600 citations, there is a surprising paucity of followup work related to their string theory claims, perhaps because their central claims were widely misunderstood. First of all, S&U contains, not one but (at least) 3 conceptually distinct derivations of black hole entropy. These include:

1. A discussion of the UV divergent entanglement entropy contribution in *semiclassical field theory*, and how it renormalizes $1/G_N$.

2. A cartoon picture of how, in string theory, the entropy comes from *open strings* ending on the horizon. This picture can be used to argue that $S/A = O(1/g_s^2) = O(1/G_N)$ in the string coupling constant $g_s$. However, this argument has not yet been made sufficiently precise to calculate the numerical coefficient (except insofar as, at the level of picture-thinking, it is equivalent to the next calculation).

3. A much more precisely defined calculation involving off-shell *closed strings* in the presence of a conical singularity. This calculation gets the factor of $1/4G_N$ exactly correct.

Unfortunately, most of the details of calculation 3 are not visible in the S&U paper, since they are "incorporated by reference" to the work of Tseytlin on off-shell string theory [5]. (This has led some people to wrongly think the S&U derivation of 1/4 is essentially circular; when in fact it has a sound basis, within an unfamiliar formalism.)

In S&U's calculation, the black hole entropy comes from the following conical variation formula [6]:

$$S = (1 - \beta\,\partial_\beta)Z_0\big|_{\beta=2\pi} \,, \tag{2}$$

where $Z_0$ is the partition function of a single spherical (genus-0) worldsheet, and $\beta$ is the total angle around the horizon. (Since there can be multiple such spheres, we have to exponentiate

to get the tree-level amplitude $Z_{\text{target}} = \exp(Z_0) = \exp(-I_0)$, where $I_0$ is the tree level effective action.)

The main conceptual subtlety of S&U arises because, on a conical manifold with $\beta \neq 2\pi$, the Einstein equations of motion are not satisfied. Hence, (2) requires a calculation in an *off-shell* string theory, where conformal invariance on the worldsheet is explicitly broken. It follows that the nonlinear sigma model (NLSM) living on the worldsheet is a QFT rather than a CFT. This means that, in addition to the Lagrangian $\mathcal{L}$, a UV cutoff $\epsilon$ with dimensions of length must be specified. The UV cutoff $\epsilon$ on the worldsheet behaves as an IR regulator in target space, so by adjusting $\epsilon$ one can make $I_0$ either approximately local, or highly nonlocal.

In part I of this work [7], we addressed this conceptual difficulty associated with breaking conformal invariance on the worldsheet using Tseytlin's NLSM off-shell formalism. We gave an accessible overview of his off-shell prescriptions, and provided a general abstract proof (using conformal perturbation theory) they give the correct tree-level S-matrix and equations of motion, at least to all orders in $g_s$ and $\alpha'$.

Although taking the worldsheet QFT off-shell requires the arbitrary specification of a Weyl frame $\omega$ on the worldsheet, we showed that at the end of the day this arbitrary choice does not matter, because the effects of changing $\omega$ can be fully absorbed into field redefinitions of the target space fields. This corresponds to renormalization of the worldsheet QFT.

In this paper (part II), we explain how Tseytlin's off-shell formalism was used by S&U to calculate black hole entropy. We will show how to explicitly use the formalism to compute the the string partition function on off-shell backgrounds, e.g. on a conical manifold.

**Plan of paper.** The outline of this paper is as follows: In section 2, we describe Tseytlin's sphere prescription, and the results in part I of this work [7] justifying its validity.

In section 3 we give a more concrete derivation of the Einstein-Hilbert action from the worldsheet theory, following the approach of [8]. It turns out that the Einstein-Hilbert term arises from the *zero mode* sector of the worldsheet. Doing this covariantly requires a careful accounting of path integral measure factors, as well as the definition of the zero mode of the $X^\mu$ coordinate field. (This is a 2-loop calculation, but as a result of some Feynman graph identities in the NLSM, it can be reduced to a 1-loop calculation, with no need to integrate over multiple momenta.) We also work out the dilaton action to the same order in $\alpha'$.

This allows us to arrive at the Susskind-Uglum calculation of black hole entropy from off-shell closed string theory in section 4. In this section, we also discuss the relationship between off-shell and on-shell black hole entropy calculations, and discuss the connection to renormalization—how RG flow smooths out a conical manifold. (We also tentatively make some first steps towards making sense of their open string picture.)

In section 5 we compare Susskind-Uglum to a rival method for calculating black hole entropy by analytically continuing (on-shell) $\mathbb{Z}_N$ orbifolds. However, the orbifold is fundamentally different from the cone because it does not allow processes in which the string pinches off at the orbifold singularity. As a result, this method does not give the correct black hole entropy—unless perhaps (following Dabholkar [9]) we allow tachyons to condense on the orbifold.

Finally, we wrap up in the Discussion 6 by suggesting possible avenues for further calculations of entropy in the off-shell formalism. We discuss the prospects for higher genus and higher $\alpha'$ calculations, as well as the bulk side of holographic AdS/CFT spacetimes, and the exact "cigar" solution.

## 2 Tseytlin's sphere prescription

This section is a brief summary of Tseytlin's sphere prescription, which we justified in the previous installment [7].

Tseytlin's off-shell NLSM formalism is a first quantized approach to string theory, in which one takes the worldsheet field theory to be a non-conformally invariant QFT. (In our work we do not need to assume that this QFT takes the form of a standard NLSM; so we can also consider highly non-geometrical string compactifications.)

On the sphere Tseytlin does *not* deal with the SL(2,$\mathbb{C}$) Möbius group by fixing 3 points, as this prescription does not properly extend to the off-shell case. Instead, at the $n$-th order of perturbation theory, he integrates *all $n$* vertex operators over the sphere to obtain a correlator $K_{0,n}$. This introduces log divergences as $n-1$ points come together on the sphere. To obtain the correct spherical string amplitude $Z_0$ for a sphere, Tseytlin therefore *differentiates* by the log of the UV cutoff $\epsilon$, so that (up to a multiplicative factor we are not bothering with) we get:

$$Z_0 = \frac{\partial}{\partial \log \epsilon} K_0 \,, \tag{T1}$$

where $K_0 = \sum_n K_{0,n}$. We call this **T1** because it was Tseytlin's *first* sphere prescription [5], and also because it involves *one* derivative with respect to the RG flow.[1]

By taking the QFT to be a nonlinear sigma model, Tseytlin checked [10–13] that this prescription gives good answers for the first few terms in the effective action $I_0$, at least for massless fields of super(string) theory in the long wavelength regime where the characteristic radius of curvature of the target spacetime $r_c \gg l_s$ [14, 15].

In [7], we justified these prescriptions with arguments that are more general than those found in Tseytlin's work. As a key lemma, we showed that when all insertions are marginal primaries, the **T1** prescription is equivalent to modding out by SL(2,$\mathbb{C}$) gauge orbits. This allowed us to recover standard string theory results from the sphere partition function, including the tree-level S-matrix and the equations of motion to all orders in perturbation theory in $n$.

As needed to go off-shell, these equations of motion are valid even for perturbations to the worldsheet action that are not marginal primaries. The precise range of validity was described more carefully in part I, but at any finite order in $n$ it includes arbitrary orders in perturbations of the operator dimension about marginality, which suffices for purposes of calculating at all orders in $\alpha'$.

Since S&U's formula for black hole entropy (2) only involves going off-shell at *linear* order ($n = 1$), and we will work at leading order in $\alpha'$, our results in part I are vastly more general than what we needed for part II. However, when trying to understand S&U we had numerous questions about what it means to take string theory off-shell, and why Tseytlin's sphere prescription **T1** can be trusted. It was only by answering all of these questions in part I, that we gained sufficient confidence that the off-shell formalism makes sense, to accept its assertions about black hole entropy. We have tried to make part II mostly self-contained for those readers who are willing to take the general validity of Tseytlin's prescription on faith. But those who wish to have the sphere prescription justified in more detail should read part I.

These results from part I provided a general abstract argument that we obtain the correct string action. But in the next section of this paper, we will get our hands dirty and explain how

---

[1]There is also a more general **T2** prescription needed to obtain the correct action for the bosonic string tachyon:

$$Z_0 = \left( \frac{\partial}{\partial \log \epsilon} + \frac{1}{2} \frac{\partial^2}{(\partial \log \epsilon)^2} \right) K_0 \,, \tag{T2}$$

but this prescription is not needed for the present paper as it is equivalent to **T1** in the regimes of interest. See part I [7] for more details.

to derive the Einstein-Hilbert action directly from the worldsheet. This will allow us finally arrive at the key result of this paper: the Susskind-Uglum calculation of black hole entropy from off-shell closed string theory, in section 4.

## 3 The zero mode of a compact worldsheet

In this section we will derive the classical Einstein-Hilbert term in the bosonic string action $I_0$ from a worldsheet perspective. For this we must consider a nonlinear sigma model on a noncompact target space manifold. In this section we calculate the target space action up to 2 derivatives, i.e. the leading order in $\alpha'$—unlike the results in chapter VI of [7] which were valid to all orders in $\alpha'$. Our analysis closely follows Tseytlin [8, 16]. In this section, we use $\epsilon$ to refer to a heat kernel regulator rather than a hard disk cutoff.

In order to write the effective action as an integral over the $D$ dimensions of spacetime, it is necessary to decompose the coordinate $X^\mu$ into a zero mode $Y^\mu$ and the nonzero modes $\eta^\mu$. That way, after integrating out the $\eta^\mu$ fields, the action takes the form of

$$\int d^D Y \, \mathcal{L}_0(Y), \tag{3}$$

where $\mathcal{L}_0(Y)$ is the spacetime Lagrangian for the light string fields.

We will then show that the target space Einstein-Hilbert term originates from the zero mode on the worldsheet. More precisely, it comes from the fact that the zero modes $Y^\mu$ behave differently than the nonzero modes $\eta^\mu$, since only the latter are confined by a quadratic potential.[2]

It is essential for this program that the nonlinear sigma model be defined in a way that respects target space covariance. There are two main hazards making this tricky:

1. The most naive way of extracting Feynman rules from the action fails to be covariant when there are dynamical fields multiplying propagator terms.[3] This issue arises whenever the target space volume is not unimodular: $\sqrt{G} \neq 1$.

2. The most obvious way to define the zero mode $Y^\mu$—just average the coordinate $X^\mu$ over the worldsheet volume—fails to respect target space covariance, because it involves the word "coordinate".[4]

To deal with issue #1, we include in the partition function a measure factor which depends explicitly on $\sqrt{G}$. This ensures that covariance remains manifest (at least up to a pure scheme dependency).

With respect to issue #2, we note that (at least in the finite $\epsilon$ regime where the string action is approximately local) the non-covariant term coming from the identification of the zero mode is a pure boundary term. So it can be easily identified and dropped.

---

[2]This means that, in bosonic string theory, an analogue of this term appears at arbitrary genus g, but for the classical black hole entropy we are interested in the genus-0 sphere case. For superstrings the higher genus (g ≥ 1) contribution to the Einstein Hilbert term vanishes due to a target space supersymmetric nonrenormalization theorem.

[3]For example, if we have a single scalar field $\phi$ whose action is $I = \int d^d x \, (1 + \phi^2)(\partial \phi)^2$, naively this introduces a 4-valent vertex which renormalizes other terms in the action, and yet that can't be true because the action is field redefinition equivalent to a free action. In this case we are missing a divergent measure factor which depends on $1 + \phi^2$. In other words, the principle of "democracy of paths", whereby all histories are weighted equally in the path integral up to phases, is valid only for theories with *constant* propagators.

[4]The standard method of dealing with this problem, the *background field method* [17–19], ensures covariant answers but introduces some additional extra complications.

To see why this is true, suppose we integrate the Lagrangian over some region $\mathfrak{R}$ which is large compared to the nonlocality scale:

$$I_0[\mathfrak{R}] = \int_{\mathfrak{R}} \mathrm{d}^D Y \, \mathcal{L}_0(Y). \tag{4}$$

In the interior of $\mathfrak{R}$, we are now integrating over both the zero modes *and* nonzero modes, without distinction. So in the deep interior, it doesn't really matter how the zero mode is defined. The only problem arises near the boundary $\partial \mathfrak{R}$, where there is an ambiguity concerning which worldsheets—remember these are extended objects!—should be counted as being "inside" or "outside" of $\mathfrak{R}$. This is really a purely conventional question, not an objective physical fact.

The zero mode $Y^\mu$ as defined above answers this question, albeit in a non-covariant manner that depends on the particular choice of coordinate system.[5] Yet because the action is approximately local, this problem can only affect the string worldsheets near the boundary. So in an $\alpha'$ expansion, the noncovariance must take the form of an integral over $\partial \mathfrak{R}$. Hence it manifests as a *total derivative* in the spacetime Lagrangian $\mathcal{L}_0(Y)$.[6]

This means that the noncovariant terms will not affect the equations of motion. We do, however, have to drop them in order to obtain the correct result for the conical entropy, since it is difficult to find a coordinate system where their effects would cancel.

As for covariant boundary terms, we cannot determine them by our current formalism. However, they cannot affect off-shell computations of black hole entropy, since any such boundary terms will be linear in $\beta$ and hence will cancel in the variation (2). This is true even at higher orders in $\alpha'$.[7] (They would, however, play a crucial role in obtaining the correct black hole entropy by an *on-shell* $\beta$ variation, as we will discuss in section 4.2.)

Having provided these salutary warnings, we are now ready to proceed to compute the worldsheet partition function.

## 3.1 Partition function

We now calculate the partition function on a compact 2-manifold $\Sigma$. Although we are primarily interested in the sphere, until the very end all our manipulations will also be valid for the torus, as we will only use the fact that the metric $g_{ab}$ on $\Sigma$ is homogeneous and has a 180° rotational isotropy.

We start with the following bare NLSM partition function:

$$Z^{(B)} = \int [\mathrm{d}X] \exp(-I_{\mathrm{QFT}}[X]), \tag{5}$$

where

$$[\mathrm{d}X] = \prod_z \mathrm{d}X(z)\sqrt{G(X(z))}, \quad G = \det G_{\mu\nu},$$

$$I_{\mathrm{QFT}} = \frac{1}{4\pi\alpha'} \int \mathrm{d}^2 z \sqrt{g} \Big( \partial_A X^\mu \partial^A X^\nu G_{\mu\nu}(X) + \alpha' R^{(2)} \Phi(X) \Big). \tag{6}$$

---

[5]More precisely, it depends on an *affine structure* on target space. So long as we remember which affine structure we are using, we are free to pass to other coordinate systems. There are manifolds with no globally defined affine structure (e.g. $S_2$) and on such manifolds it would be necessary to divide the manifold into pieces include a boundary term on the border between pieces. In this roundabout way one would recover the covariant action on compact manifolds. But it is easier to just realize this *could* be done, and drop the offending terms.

[6]In at least some contexts, this noncovariant total derivative seems to be closely related to the Gibbons-Hawking boundary term, but we are not sure how to make this idea precise.

[7]For a relevant discussion of the boundary term in the classical string field theory as well as low-energy effective actions, see the nice discussion on p.7-8 in [20].

In (5), $G_{\mu\nu}(X)$ is the spacetime background metric, $\Phi(X)$ is the dilaton. For simplicity, we don't include the antisymmetric $B_{\mu\nu}$ field.[8] The form of the path integral measure guarantees the path integral is spacetime reparametrization-invariant under the simultaneous transformation of $X$ and $G$:

$$X^\mu \to X'^\mu(X^\mu), \quad G \to G'_{\mu\nu} = \frac{\partial X^\alpha}{\partial X'^\mu}\frac{\partial X^\beta}{\partial X'^\nu}G_{\alpha\beta}. \tag{7}$$

To make the path integral well defined, we need a UV cutoff $\epsilon$. In this section we use the heat kernel regularization method to consistently cutoff the action *and* the measure. This is done by inserting a factor of $e^{\epsilon^2\Delta}$ into divergent expressions, with $\Delta = -\nabla^2$.

Let us first focus on the measure factor $[dX]$ in (5). We regulate this expression as follows:

$$Z_{\mathcal{M}} := [dX] = \prod_z dX(z)\, e^{\mathcal{M}},$$
$$\mathcal{M} = \frac{1}{2}\operatorname{tr}(\ln G \exp(-\epsilon^2\Delta))$$
$$= \frac{1}{2}\int d^2z\sqrt{g}\ln G(X(z))K(z,z;\epsilon) \tag{8}$$
$$= \frac{1}{2}N\log G(X),$$

where $K(z,z;\epsilon)$ is the trace of the heat kernel with an infinitesimal Schwinger proper time $\epsilon^2 \to 0$ (which is a regularization of the delta function $\delta^{(0)}$)

$$K(z,z';\epsilon) = \langle z|\exp(-\epsilon^2\Delta)|z'\rangle,$$
$$\lim_{\epsilon\to 0}K(z,z';\epsilon) = \delta^{(2)}(z,z') = (1/\sqrt{g})\delta^{(2)}(z-z'). \tag{9}$$

The trace is given by the heat kernel asymptotic [21]:

$$K(z,z;\epsilon) = \frac{1}{4\pi\epsilon^2} + \frac{1}{24\pi}R^{(2)} + O(\epsilon^2), \tag{10}$$

so that after regularization the effective number of modes $N$ is given by

$$N = \int d^2z\sqrt{g}K(z,z;\epsilon) = \frac{V}{4\pi\epsilon^2} + \frac{1}{6}\chi + O(\epsilon^2), \tag{11}$$

where $V$ is the volume of the worldsheet, and $\chi$ is its Euler characteristic.[9]

Now, $Z_{\mathcal{M}}$ can be written as

$$Z_{\mathcal{M}} = (\sqrt{G})^N. \tag{12}$$

This would certainly be a covariant measure for a lattice field theory with $N$ points, since each $X$ variable would be integrated with the covariant measure $d^D X\sqrt{G}$. As we are using the heat kernel regulator, the covariance could potentially be disrupted by scheme dependencies, but as we shall see in section 3.3, the fact that we are using $\epsilon$ to regulate both the action and the measure, will result in the two terms combining to give a covariant final result.

---

[8]As pointed put in [8], a choice of a local path integral measure is equivalent to a choice of the *bare* values of the tachyon and dilaton fields. In addition, the *renormalized* value of $T(X)$ may be consistently tuned to be zero because it is associated with a power law divergence.

[9]We note in passing that this formula implies that the CFT operator conjugate to $\sqrt{G}$ has an anomalous dependence on the curvature $R$ when acting with a conformal transformation. This operator is not simply $:\partial_A X^\mu \partial^A X_\mu:$, because $G$ also appears in the measure. Both of these terms contribute to the aforementioned anomaly, in order to give rise to the covariant answer required by section VII in [7].

## 3.2 Mode decomposition

The eigenfunctions $\varphi_m$ and eigenvalues $\lambda_m$ of the Laplacian $\Delta$ on a compact 2-surface of the string worldsheet are defined by the following set of relations for modes on the worldsheet:

$$
\begin{aligned}
&\Delta \varphi_m = \lambda_m \varphi_m \,, \\
&\int \mathrm{d}^2 z \sqrt{g} \, \varphi_m \varphi_n = \delta_{mn} \,, \\
&\int \mathrm{d}^2 z \sqrt{g} \, \varphi_m(z) = 0 \,, \quad m \neq 0 \,, \\
&\varphi_0 = 1/\sqrt{V} \,, \quad \lambda_0 = 0 \,.
\end{aligned}
\tag{13}
$$

We now consider the regularized Green's function defined in terms of $\varphi_n$ and $\lambda_n$ as:

$$
\begin{aligned}
\mathcal{D}(z, z') &= \left\langle z \left| \Delta^{-1} \exp(-\epsilon^2 \Delta) \right| z' \right\rangle \\
&= \sum_{m \neq 0} \frac{\exp\left(-\epsilon^2 \lambda_m\right)}{\lambda_m} \varphi_m(z) \varphi_m(z') \,.
\end{aligned}
\tag{14}
$$

Here we have omitted the zero mode ($\phi_0$)—which is good because otherwise Gauss' law prevents us from inverting the propagator on a compact worldsheet! (This is justified by the fact that we will be using this expression in Feynman diagrams that integrate over the nonzero modes only.) From (14) we obtain the important relation:

$$
\Delta \mathcal{D}(z, z') = \delta^{(2)}(z, z') - 1/V \,.
\tag{15}
$$

To compute the regulated partition function $Z_B$, we now split $X^\mu$ into a constant part and a non-constant part $X^\mu = Y^\mu + \eta^\mu$. To do this properly, and avoid over-counting of $Y^\mu$, following the standard Fadeev-Popov (FP) procedure, we insert the following "1" factor into (5) [17].[10]

$$
\begin{aligned}
&1 = \int \mathrm{d}^D Y \int \prod_z \mathrm{d}\eta(z) \delta^{(D)}(X(z) - Y - \eta(z)) \delta^{(D)}(P^\mu[Y, \eta]) Q[Y, \eta] \,, \\
&Q = \det\left(\partial P^\mu[Y - a, \eta + a]/\partial a^\nu\right)_{a=0} \,.
\end{aligned}
\tag{16}
$$

A canonical choice of $P^\mu$ and the FP factor $Q$ is

$$
\begin{aligned}
P^\mu &= \int \mathrm{d}^2 z \sqrt{g} \, \eta^\mu(z) \,, \\
Q &= V^D \,, \quad V = \int \mathrm{d}^2 z \sqrt{g} \,.
\end{aligned}
\tag{17}
$$

If we take the worldsheet to be a unit sphere, this just contributes a multiplicative constant to the partition function.

## 3.3 Covariance of the measure

To ensure the manifest covariance of the path integral measure, the noncovariant terms $\ln G(X(z))$ in (8) must cancel with some term in the path integral over $\eta$ in the action which has the number of *non-zero* modes $N'$. Let us how this happens.

---

[10]The $P^\mu = 0$ gauge conditions guarantees that the integral over $\eta$ is can be expressed only in terms of the non-zero modes of $\Delta$ by virtue of (13).

If we substitute $X^\mu = Y^\mu + \eta^\mu$ into (5) and (8) and expand, we obtain

$$
\begin{aligned}
I_B = \frac{1}{4\pi\alpha'} \int d^2z \sqrt{g} \Big( &\partial^a \eta^\mu \partial_a \eta^\nu G_{\mu\nu}(Y) + \partial^a \eta^\mu \partial_a \eta^\nu \eta^\lambda \partial_\lambda G_{\mu\nu}(Y) \\
&+ \frac{1}{2} \partial^a \eta^\mu \partial_a \eta^\nu \eta^\lambda \eta^\rho \partial_\lambda \partial_\rho G_{\mu\nu}(Y) + O(\eta^5) \Big) \\
+ \chi \Phi + \frac{1}{8\pi\alpha'} \int d^2z \sqrt{g} \Big( &\eta^\mu \eta^\nu \partial_\mu \partial_\nu \Phi(Y) + O(\eta) \Big).
\end{aligned}
\tag{18}
$$

The leading order in $\alpha'$ contribution to $Z_{\mathscr{M}}$ (focusing only on the zero mode and ignoring the $O(\eta)$ perturbations) is:

$$
Z_{\mathscr{M}}^{(0)} = \int d^D Y \exp(-\chi\Phi) \sqrt{G}^{(N)}.
\tag{19}
$$

The leading order term from the path integral over $\eta$ in (18) is

$$
\begin{aligned}
Z_\eta^{(0)} &= \left[ \det' G_{\mu\nu}(Y) \Delta \right]^{-1/2} \\
&= \exp\left[ -\frac{1}{2} N' \log G(Y) - \frac{1}{2} D \ln \det' \Delta \right] \\
&= Z_f^{(0)} \exp\left( -\frac{1}{2} N' \log G(Y) \right) \\
&= Z_f^{(0)} \sqrt{G(Y)}^{-N'},
\end{aligned}
\tag{20}
$$

where $Z_f^{(0)}$ is the l-loop free particle vacuum functional given by

$$
Z_f^{(0)} = \exp\left( -\frac{D}{2} \log \det' \Delta \right),
\tag{21}
$$

and $\det' \Delta$ includes *only* the non-zero modes of $\Delta$:

$$
\begin{aligned}
N' &= \int d^2z \sqrt{g} K'(z, z, \epsilon) \\
&= \int d^2z \sqrt{g} K(z, z, \epsilon) - 1/V \\
&= N - 1.
\end{aligned}
\tag{22}
$$

Putting $Z_{\mathscr{M}}$ and $Z_\eta$ together, we obtain a manifestly covariant measure:

$$
\begin{aligned}
Z_{\mathscr{M}}^{(0)} Z_\eta^{(0)} &= Z_f^{(0)} \int d^D Y \exp(-\chi\Phi) \sqrt{G(Y)}^{(N-N')} \\
&= Z_f^{(0)} \int d^D Y \sqrt{G} \exp(-\chi\Phi).
\end{aligned}
\tag{23}
$$

If we now include the $O(\eta^2)$ terms in the expansion of (8), then the $O(\alpha')$ correction to the target space Lagrangian from the measure factor is:

$$
Z_{\mathscr{M}}^{(1)} = \left( 1 + \frac{1}{2} \pi \alpha' N D(z, z) \left( G^{\mu\nu} G^{\lambda\rho} \partial_\lambda \partial_\rho G_{\mu\nu} - G^{\mu\alpha} G^{\nu\beta} G^{\rho\lambda} \partial_\rho G_{\mu\nu} \partial_\lambda G_{\alpha\beta} \right) \right).
\tag{24}
$$

The path integral can now expressed as the product of several factors:

$$
Z^{(B)} = Z_f^{(0)} \int d^D Y \sqrt{G} \exp(-\chi\Phi) Z_D^{(1)} Z_G^{(1)} Z_{\mathscr{M}}^{(1)}(Y),
\tag{25}
$$

where $Z_D^{(1)}$ and $Z_G^{(1)}$ are the 1-loop dilaton and 2-loop graviton multiplicative corrections, respectively which we compute separately in the next two subsections.

## 3.4 Dilaton contribution

Now we turn to the dilaton contribution. From (18), the dependence of the Lagrangian on the dilaton, expanded to $O(\alpha'^2)$ is given by

$$
\begin{aligned}
Z_D^{(1)} &= 1 - \alpha' \chi \, \eta^\mu \eta^\nu \partial_\mu \partial_\nu \Phi(Y) \\
&= 1 - \alpha' \chi \, G^{\mu\nu} \partial_\mu \partial_\nu \Phi(Y) \mathcal{D}(z,z) \\
&= 1 + \alpha' \frac{\chi}{2} (\log \epsilon + h) \, G^{\mu\nu} \partial_\mu \partial_\nu \Phi \,,
\end{aligned}
\tag{26}
$$

where in the second step, we used that the regularized propagator for $\eta^\mu$ given by

$$
\left\langle \eta^\mu(z) \eta^\nu\left(z'\right) \right\rangle = 2\pi\alpha' G^{\mu\nu}(Y) \mathcal{D}\left(z,z'\right) \,,
\tag{27}
$$

and that in the limit $z \to z'$, (14) is given by

$$
\mathcal{D}(z,z) = -\frac{1}{2\pi} \log \epsilon + h \,,
\tag{28}
$$

where on a homogeneous worldsheet (which is possible for either the sphere or the torus) $h = \frac{1}{2} \log V + O(1)$ and is independent of position. Homogeneity also ensures that there can be no tadpoles[11] of the $\eta^\mu$ field; since by symmetry, any such tadpole would be proportional to the nonexistent zero mode of $\eta^\mu$. The $O(1)$ parameter is just a constant which depends on the finite part of the heat kernel—for a torus this is a function of the modular parameter $\tau$.

## 3.5 Graviton contribution

We now turn our attention to the contribution $Z_G$ from the metric perturbation in (25). From (5), we examine the two possible 2-loop diagrams in $Z_G$ expanded to $O(\alpha'^2)$:

$$
Z_G^{(1)} = 1 + J_1 G^{\mu\nu} G^{\lambda\rho} \partial_\lambda \partial_\rho G_{\mu\nu} J_2 G^{\mu\alpha} G^{\nu\beta} G^{\rho\lambda} \partial_\rho G_{\mu\nu} \partial_\lambda G_{\alpha\beta} + J_3 G^{\mu\lambda} G^{\nu\beta} G^{\rho\alpha} \partial_\rho G_{\mu\nu} \partial_\lambda G_{\alpha\beta} \,,
\tag{29}
$$

where the corresponding Feynman diagrams (shown in Fig. 1) evaluate to:

$$
\begin{aligned}
J_1 &= \frac{1}{2} \pi\alpha' \int \mathrm{d}^2 z \sqrt{g} \left[ \left( \partial_A \partial^A \mathcal{D}(z,z) \right)_{z=z'} \mathcal{D}(z,z) \right] \\
&= -\frac{1}{2} \pi\alpha' \mathcal{D}(z,z) N' \\
&= \frac{1}{4} \alpha' N' (\log \epsilon + h) \,,
\end{aligned}
\tag{30}
$$

$$
\begin{aligned}
J_2 &= \frac{1}{2} \pi\alpha' \int \mathrm{d}^2 z \sqrt{g} \int \mathrm{d}^2 z' \sqrt{g'} \partial_A \partial_B' \mathcal{D}\left(z,z'\right) \partial^A \mathcal{D}\left(z,z'\right) \partial'^B \mathcal{D}\left(z,z'\right) \\
&= \frac{1}{2} \pi\alpha' \mathcal{D}(z,z) N' \\
&= -\frac{1}{4} \alpha' N' (\log \epsilon + h) \,,
\end{aligned}
\tag{31}
$$

and

$$
\begin{aligned}
J_3 &= \pi\alpha' \int \mathrm{d}^2 z \sqrt{g} \int \mathrm{d}^2 z' \sqrt{g'} \partial^A \partial'^B \mathcal{D}(z,z') \partial_A \partial_B' \mathcal{D}\left(z,z'\right) \mathcal{D}\left(z,z'\right) \\
&= -\frac{1}{2} \pi\alpha' \mathcal{D}(z,z) \\
&= \frac{1}{4} \alpha' (\ln \epsilon + h) \,.
\end{aligned}
\tag{32}
$$

---

[11]Here we mean the ordinary QFT tadpole diagrams like O— of the fundamental field of the NLSM, *not* the string field tadpoles discussed in other parts of this article.

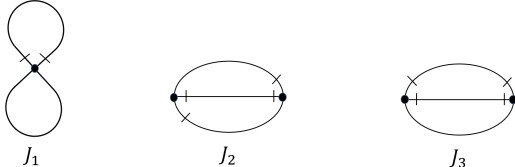

Figure 1: The three 2-loop Feynman diagrams, $J_1$, $J_2$, and $J_3$ which contribute to $Z_G^{(1)}$. The edges are the propagator of the nonzero modes $\eta^\mu$ (contracted with the metric $G_{\mu\nu}$) while the tick marks represent $\partial_A$ derivatives (always contracted with another derivative on the same vertex). See Fig. 2 for their evaluation.

Happily, we didn't actually have to do any 2-loop integrals, as all of these diagrams can be reduced to 1-loop diagrams by integrating by parts, and (in the case of the oyster diagrams $J_2$ and $J_3$) using (15) to remove propagator edges (see Fig. 2). To evaluate the resulting 1 loop expressions, we use (11), (22) and (28).

We have also used the existence of a 180° rotational symmetry on the sphere or torus to discard any loop from a vertex to itself containing a single derivative:

$$\partial_A \mathcal{D}(z,z')\Big|_{z=z'} = 0\,, \tag{33}$$

as this is odd with respect to the 180° rotation.

## 3.6 Target space effective action

Putting equations (24), (26), and (29) together, the bare path integral (25) of the nonlinear sigma model action (5) comes to

$$
\begin{aligned}
Z^{(B)} = Z_f^{(0)} \int \mathrm{d}^D Y \sqrt{G} \exp(-\chi\Phi)\Big[ & 1 + \frac{\chi}{2}\alpha'(\log\epsilon + h)G^{\mu\nu}\partial_\mu\partial_\nu\Phi \\
& - \frac{1}{4}\alpha'(N-N')(\log\epsilon + h)\,G^{\mu\nu}G^{\lambda\rho}\partial_\lambda\partial_\rho G_{\mu\nu} \\
& + \frac{1}{8}\alpha'(N-N')(\log\epsilon + h)\,G^{\mu\alpha}G^{\nu\beta}G^{\rho\lambda}\partial_\rho G_{\mu\nu}\partial_\lambda G_{\alpha\beta} \\
& + \frac{1}{4}\alpha'(\log\epsilon + h)\,G^{\mu\lambda}G^{\nu\beta}G^{\rho\alpha}\partial_\rho G_{\mu\nu}\partial_\lambda G_{\alpha\beta} + O(\alpha'^2)\Big].
\end{aligned}
\tag{34}
$$

Using the following identity for the Ricci scalar $R$ in (34):

$$
\begin{aligned}
\int \mathrm{d}^D Y \sqrt{G}\exp(-\chi\Phi)R = \frac{1}{4}\int & \mathrm{d}^D Y \sqrt{G}\exp(-\chi\Phi) \\
\times\, G^{\mu\alpha}G^{\nu\beta}G^{\rho\lambda}\big( & \partial_\rho G_{\nu\beta}\partial_\lambda G_{\mu\alpha} - \partial_\rho G_{\beta\alpha}\partial_\lambda G_{\mu\nu} \\
& - 2(\partial_\rho G_{\nu\beta}\partial_\mu G_{\alpha\lambda} - \partial_\rho G_{\beta\alpha}\partial_\nu G_{\mu\lambda})\big),
\end{aligned}
\tag{35}
$$

and after accounting for total derivative terms which we show next, we obtain the following string partition function for the sphere or torus:

$$Z_0 = Z_f^{(0)} \int \mathrm{d}^D Y \sqrt{G}\exp(-\chi\Phi)\Big(1 + \frac{1}{2}\alpha'(\log(\epsilon) + h)(R + \chi^2\partial_\mu\Phi\partial^\mu\Phi) + O(\alpha'^2)\Big). \tag{36}$$

Some comments are in order. (1) The power law divergences in (18) canceled with the choice of the local measure in (24) such that the final expression of the sphere partition function in (36) has only logarithmic divergences. The key observation however is that the origin

Figure 2: The evaluation of all Feynman diagrams required to determine the classical action $I_0$ at leading order in $\alpha'$. We treat symmetry factors, and the $-1$ for each vertex, as numerical coefficients rather than as part of the diagram values. (a) The basic manipulation rules: integration by parts, the evaluation of edges involving the Laplacian $\nabla^2$ using (15) or (11), the evaluation of a basic loop with no derivatives (28), and integrating a vertex over the worldsheet volume $V$. The notation $/\!|\!\backslash$ represents an arbitrary number of additional edges coming out of a vertex. (b) The evaluation of 1-loop and figure 8 diagrams. (c) The evaluation of oyster diagrams.

of the logarithmic divergences in (36) is the *zero mode* of the Laplacian, coming from $N-N'$ in the measure and action. (2) In calculating (36), we used the non-explicitly covariant expansion of the action (18). However, the explicitly covariant Riemann normal coordinates can be used to obtain (36) [8]. The end result is the same except for the absence of the noncovariant total derivative.

**Total derivative terms.** In order to obtain the integrand of (36), we needed to subtract off two noncovariant and one covariant total derivative terms. The noncovariant terms are:

$$\int d^D Y \, \partial_\lambda \big[ \sqrt{G} \exp(-\chi\Phi) G^{\mu\nu} G^{\lambda\rho} \partial_\rho G_{\mu\nu} \big], \tag{37}$$

and its cousin:

$$\int d^D Y \, \partial_\lambda \big[ \sqrt{G} \exp(-\chi\Phi) G^{\mu\nu} G^{\lambda\rho} \partial_\mu G_{\nu\rho} \big]. \tag{38}$$

The covariant total derivative term is

$$\int d^D Y \, \partial_\mu \big[ \sqrt{G} \exp(-\chi\Phi) G^{\mu\nu} \partial_\nu \Phi \big]. \tag{39}$$

This term allows us to express the dilaton kinetic term in the action either as $\chi \partial_\mu \partial^\mu \Phi$ or $-\chi^2 \partial_\mu \Phi \partial^\mu \Phi$; in Eq. (36) we chose the latter expression. Both terms vanish on the torus

($\chi = 0$) since $R = 0$ in our flat choice of Weyl frame. (But as the $\partial_\mu \partial^\mu \Phi$ term is a total derivative term when $\chi = 0$, its coefficient cannot actually be determined by this calculation.)

**Genus-0 effective action.** To obtain the tree-level (classical) *finite* effective action for closed bosonic strings, we first note that the spherical worldsheet correlator $K_0$ (defined in section VII.A of [7]) is exactly the same as the sphere partition function except for the inclusion of ghosts:

$$K_0 = Z_0 Z_{\text{ghosts}}. \tag{40}$$

To get the classical string action, we now simply apply the **T1** prescription—whose use we have abundantly justified in sections V–VII of [7]—and obtain:

$$
\begin{aligned}
I_0 &= -\left( \frac{\partial}{\partial \log \epsilon} K_0 \right) \\
&\propto -\frac{\alpha'}{g^2} \int \mathrm{d}^{26} Y \sqrt{G} \exp(-2\Phi)(R + 2\nabla^2 \Phi).
\end{aligned}
\tag{41}
$$

In this equation, we have assumed $D = 26$ so that the ghosts cancel out the $\log \epsilon$ in $Z_f^{(0)}$[12] and set $\chi = 2$. Also note that (41) uses the covariant derivative for the kinetic term of the dilaton.

Note that there are no powers of $\log \epsilon$ remaining in (41), so our result for the sphere is actually independent of the RG scale at this order in $\alpha'$.[13]

**Genus-1 effective action.** If we instead consider the torus, we do not differentiate by $\log \epsilon$, so our result depends on the Weyl frame when the background is off-shell. This ambiguity would need to be absorbed into an $O(g^2)$ field redefinition of the target space fields using the *tree-level* equations of motion. For a classically on-shell background, and for order unity $\tau$, the genus-1 correction takes the very simple form of a volume integral:

$$K_1(\tau) \sim \alpha' \int \mathrm{d}^{26} Y \sqrt{G}. \tag{42}$$

However, to obtain $Z_1$ we also need to integrate over the modular parameter $\tau$ (divided by $\text{Vol(CKG)} = \text{Re}(\tau)$). If we allow large $\tau$ in this integral, our perturbation theory in $\eta$ breaks down. In this regime the torus needs to be treated as an extended worldline and so the effective action $I_1^{\text{eff}}$ is no longer approximately local. In the case of bosonic strings $I_1^{\text{eff}}$ is also IR divergent due to the tachyon.

On the other hand, in superstring theory, a target space nonrenormalization theorem [22] (cf. subsection 12.6 in [23]) implies that $A_{g,n} = 0$ for $g \geq 1$, $n \leq 3$ when expanding around flat space, so in this case there is no genus-1 correction to the target space Einstein-Hilbert term.

## 4 Susskind and Uglum revisited

### 4.1 The induced gravity scenario

Now that we have argued for the validity of Tseytlin's off-shell prescriptions, in this section, we explain how Tseytlin's off-shell formalism was actually used by S&U [4] to calculate the

---

[12]Otherwise there would be a leading term proportional to $D - 26$ in the action.

[13]If we had considered higher orders in $\alpha'$, there would remain powers of $\log \epsilon$ in $I_0$, in which case the interpretation would depend on the choice of renormalization regime as discussed in section IV.C of [7]. If we want an approximately local effective action we can simply choose a finite value for $\epsilon$ (which is equivalent to removing the divergences with counterterms), and different choices of $\epsilon$ or RG scheme will be equivalent via field redefinitions. On the other hand, for want the S-matrix regime we would reinterpret these higher powers of $\log \epsilon$ as poles, as discussed in section V of [7].

tree-level BH entropy.

S&U [4] pointed out that string theory is actually an *induced* theory of gravity, in the sense that the target space manifold $\mathfrak{M}$ has no inherent action of its own, apart from the action induced by the string worldsheets that propagate on $\mathfrak{M}$. That is, we do *not* couple strings to gravity by writing down something like:

$$I_{\text{target}} = \int_{\mathfrak{M}} d^D X \sqrt{-G} \frac{R}{16\pi G_N} + I_{\text{strings}}, \tag{43}$$

but rather the graviton and its gravitational action arise *entirely* from integrating out the string worldsheets (as we did in section 3). In other words in the fundamental description there is only $I_{\text{strings}}$ and the bare value of $1/G_N = 0$, and it is only in the effective theory where there is a nonzero $1/G_N$ (and similarly various higher curvature terms in $\alpha'$ expansion).

This is morally similar to Sakharov's induced gravity proposal [24] in which $1/G_N$ is induced by 1-loop QFT diagrams which are assumed to be cut off by some unknown quantum gravity physics at the scale of the Planck length $l_p$. In fact string theory is even better, because it is already finite, having an objective UV cutoff within it because the behavior of strings smooths everything out at the scale of the string length $l_s$. Even at weak coupling where $l_s \gg l_p$, we still obtain an effective Newton constant of size

$$G_N \sim (l_p)^{D-2} \sim g_s^2 (l_s)^{D-2}, \tag{44}$$

because the tree level sphere diagrams come with a large factor $1/g_s^2$ in front. This is an effect which has no analogue in ordinary QFT.

Now what are the implications of this for black hole thermodynamics? Recall that a black hole coupled to a QFT has a *generalized entropy* [1, 2] equal to

$$S_{\text{gen}} = \left\langle \frac{A}{4G_N} \right\rangle + S_{\text{out}}, \tag{45}$$

where $A/4G_N$ is the Bekenstein-Hawking horizon entropy,[14] and $S_{\text{out}} = \text{Tr}(\rho \log \rho^{-1})$ is the von Neumann entropy of quantum fields outside of the horizon.

However, as pointed out by S&U and Jacobson [28], in an induced gravity scenario, there is *no* bare $1/G_N$ and hence *no* intrinsic horizon entropy. Instead, we have at the fundamental level

$$S_{\text{gen}} = S_{\text{out}}. \tag{46}$$

In other words, the Sakharov induced gravity hypothesis is equivalent to the statement that black hole entropy is entirely due to the entanglement of matter fields. See [29–31] for further discussion of this point.

In this section, we show that this is indeed true for string theory if we interpret $S_{\text{out}}$ as being

$$S_{\text{out}} = (1 - \beta \, \partial_\beta) Z_0, \tag{47}$$

where $Z_0$ is the partition function of the sphere worldsheet on a cone with opening angle $\beta$. In other words, $S_{\text{BH}}$ can be interpreted as $S_{\text{out}}$ in the sense that the effective field theory of classical strings which lives on the 2D cone in target space, induces the effective Newton constant from the spherical worldsheet partition function.

There are, however, some significant caveats in the above statement. First of all, in order to really interpret $S_{\text{out}}$ as a manifestly statistical entanglement entropy, there would have to be a way to factorize the Hilbert space of string theory, so as to count the states on just one side of

---

[14]We need the expectation value since the area is now an operator that depends on the gravitational backreaction of the quantum fields [25–27].

the horizon. S&U interpret these hypothetical states as being *open* strings which begin and end on the horizon. (These are really *closed* strings on the full manifold $\mathfrak{M}$, but the horizon cuts them off.) We will discuss this putative open string picture (and the problems with making it precise) in section 4.5.

Secondly, although string theory is induced in the sense that there is no fundamental *target space* Einstein-Hilbert term, there is still a fundamental Einstein-Hilbert term on the *worldsheet*. It is this which produces the $1/g_s^2$ factor in the string theory, which manifests in the open string picture as a factor of $1/g_s$ for each string endpoint (almost like a Chan-Patton factor but with a continuous range of values). It is not yet clear whether this term can be given a statistical interpretation even in the open string perspective.

In the remainder of this section, we first give an overview in 4.2 of the two equivalent approaches (on-shell and off-shell) to computing black hole and entanglement entropy in QFT and string theory. Then we present the S&U off-shell *closed string* entropy calculation in section 4.3, which implements the off-shell approach in string theory.

## 4.2 On-shell vs off-shell thermodynamics

Before we get to the S&U computation of the black hole entropy, we briefly discuss the two different methods of calculating it in semiclassical gravity and string theory:[15] (1) on-shell and (2) off-shell. To make the discussion clear and simple, let us focus on the Einstein-Hilbert (EH) action with the Gibbons-Hawking (GH) boundary term

$$S = \frac{1}{16\pi G} \int_{\mathfrak{M}} R + \frac{1}{8\pi G} \int_{\partial\mathfrak{M}} K \,. \tag{48}$$

**Gravity on-shell.** In this method, the EH term vanishes on-shell i.e. on a saddle point, and hence the entire contribution to the classical BH entropy comes from the GH boundary term

$$\ln Z_{\text{tree}} = -I_{\text{GH}} = \beta F(\beta)\,, \tag{49}$$

where $F(\beta) = -\log Z(\beta)/\beta$ is the free energy of the canonical ensemble, in terms of which the BH entropy is the computed by

$$S_{\text{BH}} = \left(\beta\,\partial_\beta - 1\right)\left(\beta F\right) = \beta^2 \partial_\beta F\,. \tag{50}$$

The on-shell method takes you to a new saddle point; in the context of black holes this means that we move to a new mass $M(\beta)$ thus changing the horizon area $A$ to first order.

**Gravity off-shell.** Here, the first order variation $\partial_\beta F$ is *independent* of the mass $M$ in the sense that the black hole geometry does not react to the variation in $\beta$ away from the equilibrium $\beta_{\text{Rindler}} = 2\pi$. This introduces a conical singularity at the black hole horizon and leads to an unstable vacuum. This is the main physical effect of introducing a conical singularity in a thermodynamic background.[16] In the off-shell method, the GH boundary term is proportional to $\beta$ and is thus irrelevant off-shell. Therefore, the entire contribution to $S_{\text{BH}}$ comes from $I_{\text{EH}}$

$$\ln Z_{\text{tree}} = -I_{\text{EH}} = \beta F(\beta)\,. \tag{51}$$

In string theory, the on-shell approach corresponds to a worldsheet theory be a CFT (supplemented with a Gibbons-Hawking like boundary term at infinity). On the other hand, the off-shell approach corresponds to taking the worldsheet to be a QFT.

In section 5, we will discuss a *different* on-shell approach involving $\mathbb{Z}_N$ orbifolds.

---

[15]The discussion in this section is largely based on Chapter 5 of [32]. See also [33].

[16]For the conical manifold to be a saddle point of $I_{\text{EH}}$ [4], there would have to be a codimension-2 membrane *source* at the tip of the cone.

## 4.3 Entropy of closed strings in Rindler

S&U compute the entropy associated with spherical worldsheets in the near-horizon region of a $D$ dimensional Schwarzschild black hole.[17] In the limit of infinite mass $M$, this can be approximated as Rindler spacetime:[18]

$$ds^2 = -\rho^2 dt^2 + d\rho^2 + \sum_{i=1}^{D-2} (\mathrm{d}X^i)^2, \tag{52}$$

which has topology $\mathbb{R}^2 \times \mathbb{R}^{D-2}$. In this limit we are neglecting subleading $\alpha'$ corrections to the black hole entropy $A/4G_N$, which would otherwise be present in an infinite series of higher curvature corrections [34, 35].

After analytic continuation to Euclidean time $\tau = -it$, the metric (52) becomes

$$ds^2 = \rho^2 d\tau^2 + d\rho^2 + \sum_{i=1}^{D-2} (\mathrm{d}X^i)^2, \tag{53}$$

which is just flat space in polar coordinates. To avoid a conical singularity at the horizon, the $\tau$-coordinate must be periodic with periodicity

$$\tau \sim \tau + 2\pi. \tag{54}$$

We can then replace the normal $\mathbb{R}^2$ with a conical manifold $\mathfrak{M}_\beta$ by simply replacing the periodicity with

$$\tau \sim \tau + \beta. \tag{55}$$

Since at the conical tip, the Ricci scalar is given by

$$\sqrt{G^{(2)}} R^{(2)} = 2(2\pi - \beta) \delta^{(2)}(X), \tag{56}$$

and hence the EH action is

$$\int_{\mathfrak{M}_\beta^{(2)}} R = 2A(2\pi - \beta), \tag{57}$$

the tree-level black hole entanglement entropy (2) can be expressed as

$$
\begin{aligned}
S_{\mathrm{BH}} &= (1 - \beta\,\partial_\beta)\left(\frac{\partial}{\partial \log \epsilon} K_0\right)\bigg|_{\beta=2\pi} \\
&= (1 - \beta\,\partial_\beta) Z_0\big|_{\beta=2\pi} \\
&= (1 - \beta\,\partial_\beta)(-I_{\mathrm{EH}})\big|_{\beta=2\pi} \\
&= (1 - \beta\,\partial_\beta)\frac{2A_\perp}{16\pi G_N}(2\pi - \beta)\bigg|_{\beta=2\pi} \\
&= \frac{A_\perp}{4G_N}\,.
\end{aligned}
\tag{58}
$$

Here $K_0$ is the renormalized sphere partition function, which is however (at this order in $\alpha'$) independent of $\epsilon$. We got the third line from 3, where we computed $Z_0$ from the closed bosonic

---

[17]We could also consider, as S&U do, the case of a KK reduced product manifold $\mathfrak{M} \times K$, where $\mathfrak{M}$ is e.g. the four dimensional Euclidean Rindler space and $K$ is a $(D-4)$-dimensional compact CFT, in which case the 4 dimensional Newton constant would be obtained from the $D$ dimensional Newton constant by dividing by the generalized volume $V(K)$. Other than this factor, the compact dimensions play no role in the argument.

[18]While the S&U paper uses type II superstring worldsheet action in $D = 10$, in this paper, we work with the bosonic string in $D = 26$. At genus-0, the Susskind-Uglum derivation is essentially identical in both cases.

sphere string worldsheet action (and showed that that the origin of the $\log \epsilon$ in $Z_0$ comes from the zero mode of the heat kernel Laplacian on the sphere). As we showed in section V of [7] that since $Z_0$ also gives the correct S-matrix as $\epsilon \to 0$, this value of $G_N$ appearing in the black hole entropy is necessarily consistent with the value of $G_N$ that would be deduced from gravitational scattering processes [36].

A word is necessary about how to justify our use of the distributional curvature formula (56) at the singular tip of the cone. Note that we cannot simply excise this singularity (replacing the topology of the target spacetime from $\mathfrak{M}_\beta$ to $S^1 \times \mathbb{R}$), because that would change the physics even at $\beta = 2\pi$ by preventing string worldsheets from crossing the codimension 2 surface. Hence we must have a way to deal with the singularity to make it regular.

In off-shell calculations of black hole entropy we usually slightly smooth out the tip over a length scale $r_*$. It turns out however that the black hole entropy is not sensitive to the value of $r_*$ because, at first order in $2\pi - \beta$, the contribution of the tip converges in the $r_* \to 0$ limit [37], and so we recover the delta function (56). On the plane, this is relatively obvious because translation symmetry of the plane $\mathbb{R}^2$ makes the position of the curvature unimportant, allowing us to freely smear it out at first order. But it the result holds more generally even on backgrounds with less symmetry.

On this smoothed out cone, we are now justified in using the target space effective action $I_0$ that was derived in section 3. We can ignore the higher $\alpha'$ corrections because their contribution to $S$ involves higher powers of the curvature which vanish on the plane.

## 4.4 RG flow of the cone

While it should be obvious to see the relationship between the entropy and the graviton tadpole, we think it's still enlightening to have it written down explicitly. Using that the graviton beta function is given by [7]

$$\beta_{\mu\nu}^{(G)} = \alpha' R_{\mu\nu} + 2\alpha' \nabla_\mu \nabla_\nu \Phi, \tag{59}$$

and (57), there is a nonzero value of the 1-point string amplitude $A_{0,1}$ due to the graviton tadpole (for a constant $\Phi$) , associated with the $\beta$ function of the metric at the tip of the cone

$$\beta_{\mu\nu}^{(G)} = \alpha' R_{\mu\nu} = \frac{\delta^{(2)}(X)}{G_N}(2\pi - \beta) G_{\mu\nu}. \tag{60}$$

So, we see that the conical deficit in target spacetime is directly related to the nonzero graviton tadpole. At $\beta_{\text{Rindler}} = 2\pi$, it vanishes.[19] This is consistent with the fact that a non-zero tadpole signals an unstable vacuum which emits strings, in this case, from the conical tip on the black hole horizon.

While it would be interesting to explore the effects of this string emission from a real-time perspective, in this section we instead explore the RG flow of the cone, due to the nonzero $\beta$ function at the tip when $\beta \neq 2\pi$. On a smooth manifold increasing the size of the cutoff $\epsilon$ corresponds to a *Ricci flow* process on the target space manifold. This statement holds only at leading order in $\alpha'$. The RG flow has additional corrections, beyond the usual notion of Ricci flow, at higher order in $\alpha'$, which can also involve the dilaton.

However, at leading order in $\alpha'$, it happens to be the case that there are no $R$ terms in the dilaton $\beta$ function $\beta^\Phi$; hence at this order, it is actually consistent to ignore the dilaton when starting with a NLSM for which it is constant. As a result, the usual Ricci flow is valid at least when working at linear order in the angle deficit $\beta - 2\pi$. This is the case that matters for (60).

---

[19]Because $\beta^\Phi$ has no dependence on curvature, there is no dilaton $\Phi$ tadpole, although there is a $\tilde{\Phi}$ tadpole due to the RG flow of the metric. See section VII in [7] for a discussion of the difference.

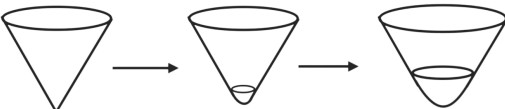

Figure 3: The apex of the cone gets smoother and smoother as we flow to the IR.

Is there a Ricci flow from a conical manifold with an arbitrary $\beta \neq 2\pi$ to $\beta_{\text{Rindler}} = 2\pi$ through which the cone becomes flat? It turns out this type of flow is known as *smoothening* Ricci flow.[20]

As usually defined, the Ricci flow describes the evolution of a *smooth* Riemannian metric on a manifold. The Ricci flow equation is then expressed in terms of a time-dependent metric $G(t)$ as

$$\frac{\partial}{\partial t} G_{\mu\nu} = -2R_{\mu\nu} =: -2(\Delta G)_{\mu\nu}, \tag{61}$$

where $R_{\mu\nu}$ is the Ricci *tensor* and $\Delta$ is a spin-2 analogue of the Laplacian operator. On a 2-dimensional conical manifold, however, it is not clear that Ricci flow even exists (in the sense that manifold will uniformize) due to the infinite (delta function) curvature at the conical singularity. Indeed, the Ricci flow equation (in the smooth part of the manifold not including the puncture), in terms of the conformal metric on the cone becomes

$$\frac{\partial}{\partial t} \omega = -2e^{-2\omega}\Delta\omega = -\frac{R}{2}. \tag{62}$$

In *conformal* coordinates, the metric on a cone can be expressed in terms of the conformal factor as[21]

$$ds^2 = e^{2(a(t)+\beta\ln\rho)}\left(d\rho^2 + \rho^2 d\theta^2\right), \tag{63}$$

where here $-1 < \beta \leq 0$ and $a(t)$ is a finite and bounded function.[22] Thus, (63) says that the information about the conical singularity is encoded in the logarithmically-divergent $\beta\ln\rho$ as $\rho \to 0$ (the asymptote of $\omega$ i.e. as we go arbitrarily close to the puncture in the center of the disk.

Including the metric asymptote $\rho \to 0$ in (62) gives an ill-defined flow equation due to the unbounded curvature at the tip.[23] Thus, to have a well-defined Ricci flow equation, the $\rho \to 0$ singular point must be truncated by putting consistent boundary and initial conditions. In this case, it was in fact shown in [38, 40] that a *unique* smoothening Ricci flow, that satisfies the flow equation for any time $t \in (0, T]$ exists on this truncated, or blunt, cone. Importantly, the curvature of the flow was found to be *bounded* at finite RG time so that the cone evolves into a smooth manifold.[24]

This means that there is a sense in which string theory automatically smooths out the cone for us. Suppose we introduce the conical singularity in target spacetime at some specific value of the UV cutoff $\epsilon$, and then we RG flow the worldsheet theory towards an IR, to a new length scale $\mu > \epsilon$. (E.g. we could fix $\mu$ to be a dimensionless number times the worldsheet sphere radius.) At the scale $\mu$, the effective off-shell theory is now that of strings propagating on a

---

[20]The definitions and discussion in this section are largely drawn from chapters 4 and 5 in [38].

[21]A conical surface is homeomorphic to a *punctured* disc with the metric (63) in the neighborhood of the puncture at the center.

[22](63) is consistent with equation 4.5 in [39].

[23]Using (56) in (62), we get $-2e^{-\beta\log\rho}\Delta\omega = 2(1+\beta)\delta^{(2)}(\rho)$, which shows the delta function singularity at the right hand side is directly related to the metric asymptote $\rho \to 0$, which by definition, in not included in the disk on the left hand side!

[24]It would be interesting to use these results to try to calculate the Rindler entropy $S(\beta)$ for $\beta \neq 2\pi$.

*smooth* background. As the Euler characteristic guarantees that

$$\int \mathrm{d}^2 X \sqrt{G^{(2)}} R^{(2)} = 2(2\pi - \beta) = \text{const.}, \tag{64}$$

in RG time, the black hole entropy $S$ remains the same at all values of $\epsilon$.[25] Note that in (64) we are defining $\beta$ as the asymptotic periodicity as $\rho \to \infty$, which is unaffected by finite quantities of RG flow.

If we take the limit that $\epsilon \to 0$ while holding $\mu$ fixed,[26] the curvature $R$ spreads out and goes to 0 at every point in the manifold. So asymptotically the Rindler cone relaxes to *flat spacetime* $\mathbb{R}^2$, in a process that takes us back to on-shell string theory asymptotically. However this limit is quite subtle as (64) still holds at every point along the flow. What is happening is that the curvature diffuses out to infinity. Hence, in order to successfully take the on-shell limit without a discontinuous jump in the action, we will need to include a boundary contribution to $S$ out near infinity, as required by the on-shell black hole entropy calculation.

If, rather than having perfect Rindler spacetime, we instead started with a black hole space-time as we did at the start of 4.3, then to take the IR limit we would need to do the Ricci flow on the Euclidean black hole instead of the plane. We expect that in this case we would similarly relax to an on-shell black hole, but at a new inverse temperature $\beta$. In this way, the off-shell string calculation is presumably equivalent to an on-shell black hole string theory calculation. But doing this calculation properly would require a better understanding than we currently possess of how the GH boundary terms are produced at the level of the worldsheet theory.

## 4.5 Towards an open string picture?

So far we have shown how Tseytlin's work on off-shell string theory was used to derive the S&U *closed* string calculation. Explaining that result was the main point of this article.

In this section—which is far more speculative—we now turn to the less rigorously defined *open* string picture, which in S&U paper was essentially based on cartoon drawings of how string worldsheets might be embedded in a geometry (see Figures 1-5 of S&U [4], and also [41] for the corresponding Feynman diagrams in the particle limit.)

The goal of the open string picture is to provide a manifestly statistical interpretation of the string entanglement entropy. The existence of such a picture is strongly suggested by the success of the closed string picture in calculating the $A/4G_N$ term even at weak coupling.

For a true statistical interpretation to exist, we need a tensor factorization of target space into two Hilbert spaces, describing strings both inside and outside the horizon:[27]

$$\mathcal{H} \subseteq \mathcal{H}_{\text{out}} \otimes \mathcal{H}_{\text{in}}. \tag{65}$$

In Lorentzian signature, these would correspond to the left and right wedges around the bifur-cation surface of the horizon. In Euclidean signature, the Hilbert space $\mathcal{H}_{\text{out}}$ would describe the state on a *ray* in $\mathbb{R}^2$ of constant $\tau$ coming out from the bifurcation point.

Since strings can cross the horizon, the description of string states in just $\mathcal{H}_{\text{out}}$ would seem to require open strings that end on the horizon, as shown in Fig. 4. This is why we wrote $\subseteq$

---

[25]Or, in the case of black hole entropy at higher orders in $\alpha'$, we would still have $\mathrm{d}S/\mathrm{d}\epsilon = 0$ since renormalization preserves the effective action $I_0$, but this would involve a more complicated computation between target space field redefinitions and the effects of changing $\epsilon$ on the sphere.

[26]Note that we are holding the coupling constants fixed at $\epsilon$. This differs from the more usual way of thinking about renormalization where we hold the physics fixed at $\mu$ and adjust the couplings as $\epsilon \to 0$. That would involve inverse Ricci flow which seems to be ill-defined when applied to the conical singularity.

[27]Ref. [42, 43] attempted to calculate an entanglement entropy in string theory by assigning string fields a position based on their center-of-mass only. This seems conceptually problematic, since the vibration of a string can cause it to partially exit a region.

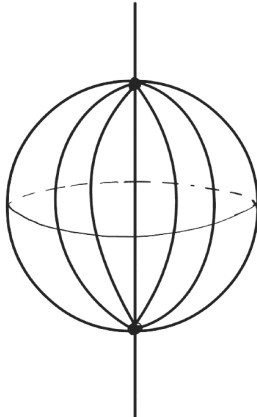

Figure 4: A *closed* spherical worldsheet is sliced vertically along constant Euclidean Rindler time. Each slice of the sphere appears to the Rindler observer as an *open* string with its endpoints frozen on the codimension-2 entangling surface (horizon). In this cartoon we assume the sphere intersects the horizon exactly twice—which is not actually realistic!

rather than $=$ in (65) because (at least in the low energy description) there seem to be edge mode constraints relating the two sides, e.g. that the number and positions[28] of the string endpoints must agree on both sides.[29] The positions of these string endpoints on the horizon are frozen, due to the infinite gravitational time dilation. So the dynamical degrees of freedom are those of an $n$-punctured sphere where $n$ is the number of intersections with the horizon.

If such a description existed, one could then write the punctured sphere partition function as a one-sided thermal ensemble:[30]

$$Z(\beta) = \mathrm{Tr}_{\mathrm{out}} \exp(-\beta K), \qquad (66)$$

where $K$ is the Killing Hamiltonian acting on states in $\mathcal{H}_{\mathrm{out}}$, which looks like a boost at the horizon. It might then be literally true[31] that the black hole entropy is a von Neumann entropy:

$$S = \mathrm{Tr}_{\mathrm{out}}(\rho \log \rho^{-1}). \qquad (67)$$

In order to ensure that there is a literal state counting interpretation, one might want to cut out a small disk $\mathfrak{D}$ around every point $p \in \Sigma \cap H$ in which the string worldsheet $\Sigma$ crosses the codimension-2 bifurcation surface of the horizon $H$. For this to work, it is crucial that the boundary conditions on $\partial\mathfrak{D}$ be chosen so that the value of $Z(\beta)$ is the same as on the original closed string worldsheet before cutting out the disks. These boundary conditions would also need to be local in the $\tau$ direction on $\partial\mathfrak{D}$, in order to ensure the validity of the Hamiltonian formalism (66). Since, on a $t = 0$ slice, the two sides of the disk are related by entanglement only, this would provide a concrete realization of the ER = EPR conjecture [45–48], in which a geometric connection is equivalent to entanglement of disconnected systems. (See [49] for other proposals for implementing ER = EPR on string theory backgrounds.)

---

[28]Although these positions fluctuate wildly so it is not totally clear how meaningful the position of a string endpoint on a compact horizon is.

[29]But see Harlow [44] for (i) an argument that there can be no fundamental edge mode degrees of freedom in a holographic theory of quantum gravity, and (ii) a toy model showing how it is possible for there to be edge modes in an effective description even though they are not present in a more fundamental description.

[30]This presumes the horizon is thermodynamically stable in the canonical ensemble, as would be true e.g. for large black holes in AdS.

[31]In the case of a black hole with finite horizon area. For a Rindler horizon there would still be annoying IR issues requiring the use of type III von Neumann algebras.

Varying $\beta$ would now be associated with varying the total angle of each circle $\partial\mathfrak{D}$. Since the Einstein-Hilbert action $\int\sqrt{g}R$ on the *worldsheet* provides a term in the effective action proportional to $2\pi-\beta$ for each disk of angle $\beta$ on the *worldsheet*, it is necessary for each disk to come with this factor. But, any local classical boundary term on $\partial\mathfrak{D}$ will produce a term linear in $\beta$ [50]. The constant term $2\pi$ must therefore come from some quantum statistical state counting. It seems to correspond to an $O(1/g_s)$ number of states associated with each endpoint, so that a string with 2 endpoints on the horizon contributes a factor of $1/g_s^2 \sim 1/G_N$ to the black hole entropy.[32]

This open string picture has already been realized in two-dimensional [51, 52] as well as six-dimensional topological string theory [53]. Whether it can be concretely realized for bosonic strings or superstrings is still an open and very challenging question.

In addition to the fact that the correct boundary conditions at $\partial\mathfrak{D}$ are unknown,[33] there is a very serious problem with making sense of these open strings. The scariest problem is that any given compact worldsheet $\Sigma$ (e.g. a sphere) will actually intersect the horizon $H$ *infinitely* many times! This is because the $X^\mu$ field on the worldsheet is actually a quantum field which, like every QFT, has violent fluctuations at short distances on the worldsheet [57,58]. Although these divergences are merely logarithmic, they still ensure that the fluctuations at any point $p \in \Sigma$ of some specific coordinate $X_0$ diverges:

$$\left\langle (X_0)^2(p) \right\rangle = \infty \, . \tag{68}$$

What's more, since UV divergences are local, if we take two distinct points $p$ and $q$ even their difference $X_0(p) - X_0(q)$ diverges wildly. See the discussion in section IV of [7] where we discuss how divergences are related to propagation of strings.

Therefore, if we are looking at the unregulated worldsheet theory $\epsilon = 0$, we cannot consistently suppose that a sphere intersects $H$ at 2 points, or even Taylor expand in the number of intersections $|\Sigma \cap H|$. Fortunately, since we already needed to introduce a UV regulator $\epsilon$ to make sense of off-shell string theory, we could choose our regulator so that it also solves this intersection problem. One way to do this would be to add a new stiffness term to the string worldsheet action which in flat spacetime would take the form:

$$\epsilon^{2n-2}\sqrt{g}X^\mu(\nabla^2)^n X_\mu \, . \tag{69}$$

Since this term is quadratic in the $X$ field, it can be viewed as a Pauli-Villars modification of the $X$ propagator:

$$\frac{1}{p^2} \to \frac{1}{p^2 + p^{2n}} \, , \tag{70}$$

where $n = 1$ is the standard propagator term, and $n \geq 3$ suffices to regulate all logarithmic and quadratic divergences on the worldsheet.

The stiffness term also ensures that the $(n-1)$st derivative of the worldsheet $\Sigma$ becomes continuous because otherwise there is an infinite penalty in the action. This appears to be strong enough to ensure that the set of intersections $\Sigma \cap H$ is finite, and that generically the string intersections take a simple form that adds $\pm 1$ to the winding number (since these sum to 0, the total number of intersections must be even.) We could then Taylor expand in the number of intersections. But we leave a detailed calculation of this proposal to future work.

---

[32]As naively the endpoints of strings are integrated over the entire volume of the horizon, it seems that the effect of finite string coupling is in some way to regulate or discretize the number of allowed string end states. However the fact that $1/g$ varies continuously suggests that things are more subtle than a simple Chern-Paton factor with $N \in \mathbb{Z}$ states running around $\partial D$.

[33]A *nonconformal* boundary state representing a disk with boundary condition $\beta \neq 2\pi$, could be described by the insertion of a vortex state, a hole, on the worldsheet. In target spacetime, the vortex is a string winding mode [54–56]. This picture seems to suggest there is an RG flow on $\partial\mathfrak{D}$ that takes the vortex to a *conformal* state $\beta = 2\pi$ where the winding tachyon condenses on the horizon, at which point, the black hole entropy is entirely the entropy of the condensate. For further discussion of this point, see section 5.2.

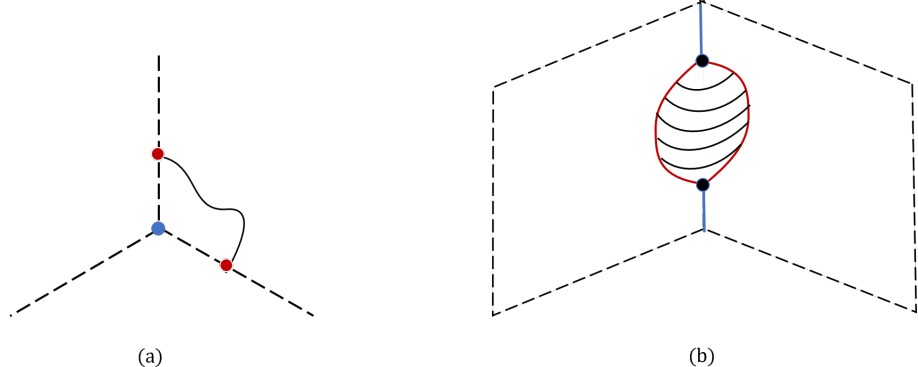

Figure 5: (a) A $\mathbb{Z}_3$ orbifold geometry is shown, where the blue dot is the location of the singularity and the dashed lines represent identified surfaces. A string with twist $k = 1$ is depicted. Despite appearances, this string is *closed* because the two red points are identified. Such twisted strings are confined near the singularity. These kinds of twisted strings also exist on a cone with $\beta = 2\pi/3$, which has the same physics except near the tip of the cone. (b) The same cone/orbifold geometry, but now the extension into some 3rd dimension (perhaps time) is depicted. A twisted spherical topology worldsheet is shown in front of the singularity (still blue). The black curves represent time slices of the twisted string, with the red curves being identified. The black dots represent a *process* in which the twisted string pinches off at the singularity. This pinching process is allowed for a smoothed out cone, but it is *not* allowed on the orbifold (prior to tachyon condensation) because the orbifold background conserves twist mod $N$. The absence of this process explains why the analytic continuation of the orbifold to arbitrary $\beta$ doesn't have a geometric interpretation, nor does it recover the Bekenstein-Hawking entropy $A/4G_N$. Tachyon condensation may alleviate this problem, since now a string can pinch off by exchanging its twist with the condensate.

## 5 Comparison to the orbifold method

We now wish to contrast the Susskind-Ulgum approach to an alternative approach [9, 59–63] to calculating string entanglement entropy, involving *orbifolds*. These are *on-shell* Euclidean string noncompact backgrounds of the form:

$$\mathcal{O} = \frac{\mathbb{C}}{\mathbb{Z}^N} \times \mathbb{R}^{D-2}, \tag{71}$$

obtained by starting with Minkowski space $\mathbb{R}^D$ and quotienting by rotations over angles that are multiples of $2\pi/N$.[34] This introduces an orbifold singularity with opening angle $\beta = 2\pi/N$. (In superstring theories, it is also necessary to take $N =$ odd in order for the boundary conditions for the fermions to be such that $\psi \to -\psi$ under a $2\pi$ rotation; the analytic continuation of the $N =$ even case does not recover the expected physics at $N = 1$.)

In addition to projecting out string states whose angular momentum is not a multiple of $N$, orbifolding also introduces a new class of *twisted states* (see Fig. 5(a)), which have winding numbers $k \in \{1, \ldots, N-1\}$, while the states inherited from the original theory have $k = 0$. This quantum number is conserved mod $N$. The ground states of these twisted sectors are *twisted tachyons*. Taking type II superstrings as an example,[35] these twisted tachyons have a

---

[34]Of course, as in the case of Susskind-Ulgum the transverse directions $\mathbb{R}^{D-2}$ could be replaced by an arbitrary string compactification.

[35]The lower bound on the tachyon dimension will be different from this in heterotic or bosonic string theories.

mass:

$$M_{TT}^2 = -\frac{4}{\alpha'}\left(1 - \frac{k}{N}\right), \tag{72}$$

which corresponds to a dimension

$$\Delta_{TT} \geq 1 + \frac{k}{N}, \tag{73}$$

which is compatible with the $\Delta > 1$ bound on worldsheet supersymmetric operators mentioned in section VI.H of [7], even though the background breaks target space supersymmetry. The GSO projection then eliminates the tachyons with $k =$ even, including the original $k = 0$ bulk superstring tachyon [64].

These twisted tachyons are reminiscent of the winding tachyons that appear in flat space compactified with a sufficiently small thermal circle $S_1$ [54–56]. However, unlike the case of $\mathbb{R}^{D-1} \times S_1$, these twisted tachyons are *localized* at the tip of the orbifold cone, because it is only there that the radius of the winding circle becomes small. (A twisted string far from the origin would have a very long length and hence a large energy.)

The orbifold construction only makes sense as a unitary string theory background for integer values of $N$. Nevertheless, because the orbifold looks awfully similar to a cone with angle $\beta = 2\pi/N$, it is tempting to regard it as if it were a thermal background with inverse temperature $\beta$, and analytically continue it towards $N = 1$, so that (analogously to (2)) the orbifold replica entropy coming from all genera g is

$$S(\rho_1) \overset{?}{=} \sum_g (1 - N\partial_N) Z_g(N)\Big|_{N=1}, \tag{74}$$

where $\rho_1$ is the Rindler state defined by $N = 1$. This orbifold replica trick was inspired by the *standard* replica trick [65,66], in which one analytically continues a $Z(\mathcal{N})$ with $\beta = 2\pi\mathcal{N}$ (which can be done even in situations without a U(1) rotational symmetry). However, in the orbifold case $N$ comes into the numerator rather than the denominator.

The evaluation of (74) depends critically on our treatment of the twisted tachyons. Most authors to propose the orbifold replica trick [61–63] take the original background before the tachyons condense, and hope that in the $N \to 1$ limit the tachyons don't matter too much. This approach suffers from a number of problems, and we believe it does not give the correct entropy at $N = 1$. In particular, this method does not give the tree-level $A/4G_N$ contribution to the entropy found by S&U.

On the other hand, in the version of the proposal defended by Dabholkar [9] (who was inspired by [39]), the tachyons are allowed to condense, and one hopes there is a minimum of the potential (which seems likely to be true by virtue of supersymmetry). We would then need to calculate black hole entropy in the *new* background, that arises as the Euclidean spacetime asymptotically settles to its new ground state under RG flow. This is a very interesting approach which plausibly would give the correct entropy, and might even help to illuminate the open string picture of Susskind and Ulgum.

We now describe these two approaches in more detail.

## 5.1 Without tachyon condensation

The first thing to note about (74) is that the integer $N$ orbifold solutions are on-shell solutions, and therefore, because the worldsheet is a CFT,[36] the genus-0 diagrams vanish (modulo a possible boundary term which will not contribute to the entropy due to being linear in $\beta$).

---

[36]Although the bulk geometry is not smooth, the worldsheet argument in section II.B of [7] will still apply. On the worldsheet, the orbifold takes the form of a discrete $\mathbb{Z}_N$ gauge field, and its sole effect on the sphere is to divide the amplitude by $N$.

In a perturbative expansion, this property will be inherited by the analytic continuation to non-integer $N$, and hence the orbifold replica trick cannot give us the leading order $A/4G_N$ contribution to black hole entropy. Instead, the first nontrivial closed string contribution starts with the genus-1 torus diagrams.

This already implies that the orbifold $\mathcal{O}$ backgrounds must be fundamentally different from off-shell conical backgrounds at the same value of $\beta$. The key difference between these two backgrounds can be seen in Fig. 5(b): the orbifold conserves twist and hence does not allow twisted strings to pinch off at the tip, while the off-shell NLSM of a slightly smoothed out cone obviously does allow this process.

On a cone of angle $\beta$, the winding number $k$ is quantized in units of $k \in \mathbb{Z}$, but it is not conserved (except mod 1 obviously). On the other hand, the orbifold conserves the twist $k$ mod $N$. When $N$ is not an integer, this conservation law fails to align with the quantization of winding modes, signalling that the analytically continued orbifold is a fundamentally non-geometrical construction.

Put another way, it is implausible that $\mathcal{O}$ can be interpreted as the thermal partition function of any unitary statistical mechanical system at inverse temperature $\beta$, since periodic partition functions only have a thermal interpretation when they can be written in terms of a time-independent Hamiltonian as

$$\exp(-\beta H) = \operatorname{Tr} \rho_1^{1/N}, \tag{75}$$

which requires there to at least be some notion of geometrical locality in the time direction.[37]

An additional problem is that the torus diagrams with genus-1 suffer from IR issues associated with the tachyon. This makes the analytic continuation of the twisted tachyon quite subtle.

For open strings on $\mathcal{O}$, a better analytic continuation behavior of $\operatorname{Tr} \rho_1^{1/N}$ was found by Witten [63], although divergences from the closed string tachyon exchange propagating down the cylinder diagram (in the crossed channel) have to be carefully handled.[38] Ref. [63] also found evidence that the analytically continued orbifold, if interpreted as a thermal partition function, does not correspond to a unitary theory.

## 5.2 After tachyon condensation

We now consider a distinct order of limits in which we *first* allows tachyons to condense at finite $N$, and only then do we take the $N \to 1$ limit.

One way to allow the tachyons to condense is to turn on a potential for twist terms in the string worldsheet. Then one can RG flow this theory in order to seek out the ground state of the system. In a supersymmetric theory one expects on positive-energy grounds that there is a stable ground state.

Adams, Polchinski, and Silverstein [39] conjectured that after RG flowing all the way to the IR limit, the orbifold relaxes to the usual flat spacetime $\mathbb{C} \times \mathbb{R}^{D-2}$ *without* the orbifolding. Inspired by this conjecture, Dabholkar [9] then showed how it might be used to calculate black hole entropy.

Specifically, [39] analyzed the RG flow in two regimes based on the relative size of the smoothed region of the cone to the string scale. In the "substringy" regime, they used D-brane probes and showed that the orbifold decays in a series of steps from $\mathbb{Z}_N$ to $\mathbb{Z}_{N-2}$, for integer $N$ until it completely flattens out. The also used the NLSM regime to study the relaxation of the cone, obtaining similar results to our section 4.4.

---

[37]However, it might still give the right answer if we restrict attention to the contribution from worldsheets which always remain far from the horizon, which plausibly includes e.g. $\log M$ corrections to black hole entropy.

[38]The 1-loop partition was found to be holomorphic in a larger region $N > 0$ and a result, analytic continuation to $\operatorname{Re} \mathcal{N} > 1$ was tachyon-free, where $\mathcal{N} = 1/N$.

In fact, in [40],[39] an exact solution of the Ricci flow equation was found that studied the decay of the orbifold (cone) $\mathbb{C}/\mathbb{Z}_N$ to another one $\mathbb{C}/\mathbb{Z}_{N'}$ with $N' < N$ (including the plane). This was done in the context of tachyon condensation. The solution exhibits the properties discussed in 4.4.

In other words, the recovery of flat spacetime seems to involve two distinct physical effects. First of all, (i) the tachyon condensation "heals" the orbifold singularity by allowing processes in which twisted states pinch off at the singularity, putting one back in the same class of off-shell backgrounds as the NLSM smoothed out cone backgrounds. But secondly, (ii), such vacua are unstable under Ricci flow. And the end state of the conical RG is just the flat vacuum!

To describe the effects of the tachyon field, Dabholkar [9] considered the following action with a twisted tachyon potential $\mathbb{V}(T)$:

$$
\begin{aligned}
-I_0 = {} & \frac{1}{16\pi G_N} \int_{\mathfrak{M}} d^D X \sqrt{G}\, e^{-2\Phi} \left[ R + 4(\nabla\Phi)^2 - \delta^{(2)}(X)\mathbb{V}(T) \right] \\
& + \frac{1}{8\pi G_N} \int_{\partial\mathfrak{M}} d^{D-1} X \sqrt{G^{(D-1)}}\, e^{-2\Phi} K .
\end{aligned}
\tag{76}
$$

Here, Dabholkar is using a convention in which the potential $\mathbb{V}$ is positive when $T = 0$, before tachyon condensation, and zero for the minimum of $\mathbb{V}$ after condensation—assuming the hypothesis is correct that the tachyon condensate is equivalent to flat space with no angle deficit.[40]

To lowest order in $\alpha'$, the equations of motion at the tip tell us that (assuming a constant dilaton):

$$
\sqrt{G^{(2)}} R^{(2)} = \left( \frac{1}{16\pi G_N} \right) \mathbb{V}(T) \delta^{(2)}(X).
\tag{77}
$$

Using the relation (56), we see that how the tachyon potential $\mathbb{V}(T)$ acts an explicit source to the conical deficit

$$
\delta = (2\pi - \beta) = 8\pi G_N \mathbb{V}(T).
\tag{78}
$$

As discussed in section 4.2, we have a choice between an on-shell or an off-shell calculation of black hole entropy. If we RG flow all the way to the IR, then that puts us back on-shell, so the contribution to the entropy $S$ would come entirely from the boundary GH term, hence we recover the Bekenstein-Hawking entropy:

$$
S = \beta^2 \partial_\beta F = -2\pi \partial_N F = \frac{A}{4G_N},
\tag{79}
$$

where the free energy is $F = (1-N)(A/8\pi G_N)$ after subtracting the flat spacetime divergent contribution.

On the other hand, if we stop the flow at a finite but large value of RG time, then we instead expect a large Gaussian-like spread of curvature.[41] We could then calculate $S$ by off-shell methods. The Gauss-Bonnet theorem would guarantee that at large RG time, the total action is linear in the asymptotic angle deficit $2\pi - \beta$, so we would still recover the Susskind-Uglum $S = A/4G_N$ answer.

Unlike the case where tachyons do not condense, it is expected that a sensible and well-behaved analytic continuation exists for $\beta$, as needed for (79). In other words, it is the order

---

of analytic continuation and tachyon condensation that matters; while attempting the former before the latter can be problematic, allowing the tachyons to first condense should avoid the analytic continuation problems.

[39]It was also demonstrated in [40] demonstrated, in asymptotically flat target spaces, that the ADM energy of target spaces with IR cutoff, is a monotonically decreasing function.

[40]If we used the opposite convention in which $\mathbb{V}(0) = 0$, and hence $\mathbb{V} < 0$ in the ground state, we would need to attribute a positive tension to the orbifold itself, which would then be cancelled by the negative tension of the tachyon condensate in its ground state.

[41]Unlike the case described in 4.4), there may also be a perturbation to the dilaton field which would similarly spread out in a Gaussian-like manner. But this does not contribute to the action at late RG time.

The idea of using tachyons as a cosmic brane source for the conical singularity is rather nice, since their existence is of special significance to string theory. However, to the best of our knowledge, the details of how to derive the twisted tachyon potential as well as the GH boundary term in (76) from the *worldsheet* are not known although an attempt to calculate $\mathbb{V}(T)$ in closed string field theory [67] was promising. Using the on-shell string action in trying to calculate $\mathbb{V}(T)$ gave nonsensical results;[42] going off-shell, on the other hand, gives more promising answers. Specifically, when truncating the closed string field theory action at cubic order, [67, 68] found a depth of the tachyon potential that was 35% of the expected potential,[43] and with recent developments in computing higher order corrections of the closed string field theory action using machine learning in [69] (based on earlier work by [70]), it may be possible to improve this result. Thus, although the tachyon potential forces the strings to be on-shell, to actually compute $\mathbb{V}(T)$ seems to require off-shell string theory.

We end by commenting on a partial relationship between the tachyon condensate and the open string picture of S&U that we discussed in section 4.5. By open-closed string duality, any process in which one absorbs a closed twisted string from the condensate, may be equivalently described as allowing additional types of processes involving open strings on the horizon. Thus, tachyon condensation on the orbifold gives a partial analogy for how the counting of open string states may arise from a more fundamental statistical description.

However, this orbifold condensate does not count as a full implementation of ER = EPR [45–49] in string theory. The reason is simply that the $\mathbb{C}/\mathbb{Z}_N$ orbifold already permits the twist $k$ to change by multiples of $N$ even *before* the condensate forms.[44] To obtain an ER = EPR picture we would instead need to start on a background in which strings are *never* allowed to cross the horizon, and then let tachyons condense on that background, so that *all* twist-changing processes result from the tachyon condensate.

For example, to explicitly exclude all twist-changing processes, we might instead start with a narrow wormhole connecting two asymptotic $\mathbb{R}^2$ regions, and then apply a $\mathbb{Z}_2$ orbifold so as to produce a non-simply connected spacetime with only one asymptotic region. See Figure 6. This would produce an off-shell spacetime with periodicity $\beta = 2\pi$ (although, as the spacetime is not simply connected, this could be adjusted to arbitrary values of $\beta$). For a sufficiently narrow wormhole, one might then expect the tachyons to condense, allowing strings to pinch off at the tip.

Although this construction is inherently off-shell, it might well RG flow to an on-shell configuration after tachyons condense. If that on-shell configuration turns out to be equivalent to the flat space string background, one would have a concrete situation in which all geometrical connection effects emerge from the behavior of entangled strings. This would be a concrete realization of ER = EPR in string theory.

The presence of tachyons should be related to a Hagedorn transition of strings in Rindler spacetime; there is evidence in the literature that this occurs at a critical temperature, the exact value of which, depends on whether the strings are bosonic, Type II or heterotic. For earlier work, see [71] and the discussion in section 3 of [61]; for more recent work, see [72]

---

[42]The on-shell action predicted about 1241% of the expected depth of the conical orbifold! We believe this may be because a proper on-shell calculation would need to drop the negative energy in the curvature/dilaton pulse noted by [39] which goes off to spatial infinity.

[43]In [67], an agreement of 72% with the predicted minimum of the potential was reported but this large agreement was found to be due to an error in identifying the orbifold gravitational coupling with the its flat space counterpart.

[44]Unless perhaps we take an $N \to \infty$ limit?

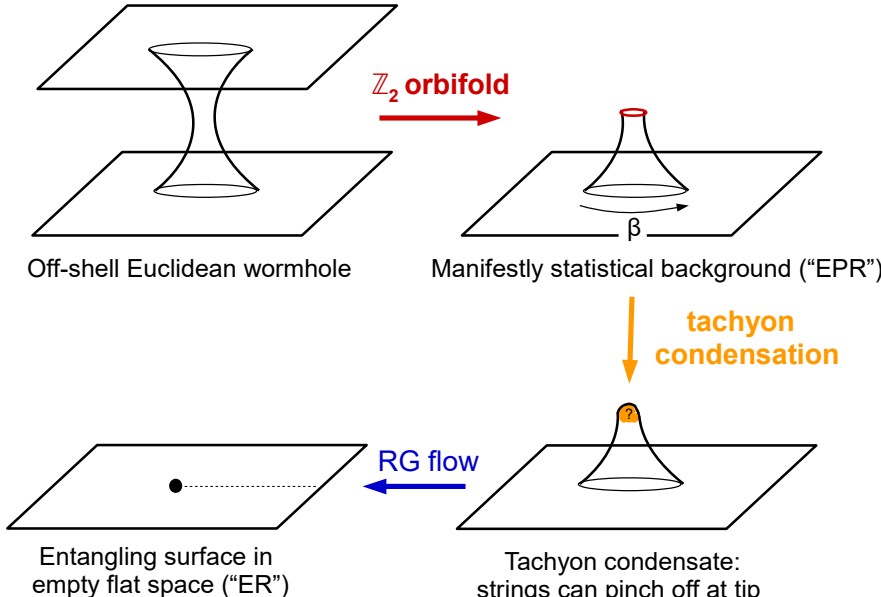

Figure 6: A proposed set of steps for implementing ER = EPR in string theory. Start with an off-shell wormhole (*upper left*), and identify the two sides with a $\mathbb{Z}_2$ orbifold. This gives an off-shell background (*upper right*) with the topology of a disk cut out around a codimension-2 surface. This picture is manifestly statistical, as there is locality around the thermal circle direction (labelled as $\beta$). Let us suppose that this background has winding mode tachyons which condense to form a new background (*lower right*), in which strings can pinch off at the tip. Plausibly, this background would RG flow towards flat spacetime (*lower left*) in the case where $\beta = 2\pi$. If we start with $\beta \neq 2\pi$, we instead expect the RG flow to converge in the IR towards the flowing cone trajectory, discussed in section 4.4.

and [32] for an extensive discussion and review of the matter.[45]

In support of the S&U open string picture, the contribution of a winding condensate to the entropy is of order $O(1/G_N) = 1/g_s^2$. Some evidence for this can be seen in the work of Horowitz and Polchinski [73], based on earlier work in [74], who found a string background, that involves a winding condensate *near* the Hagedorn inverse temperature $\beta_{\text{Hag}}$ in the form of highly excited self-gravitating oscillating strings. For recent work on this subject, see [75–77].

# 6 Discussion

## 6.1 Summary of results

The main result of this paper (part II) was to explain the underlying conceptual structure of the S&U black hole entropy argument. We showed explicitly how the effective action $I_0$ and the entropy $S = A/4G_N$ may be calculated from the sphere diagrams, in sections 3 and 4.3. We also discussed the behavior of the S&U entropy under RG flow. Although the conical manifold smooths out under RG flow, moving towards an on-shell configuration, the entropy doesn't change.

---

[45]We did not observe any Hagedorn phase transition in $\beta$ in the closed string calculation in section 4.3, but this is presumably because the closed string picture is post-tachyon condensation and therefore is stable.

We then compared these off-shell results with the (much more popular) *orbifold method* for calculating entropy from the on-shell $\mathbb{C}/Z_N$ background (5). By considering processes involving twisted string states, we concluded that the orbifold method is physically incorrect—unless one allows tachyons to condense on the orbifold, in which case it appears (though the off-shell string field theory calculations are difficult and we did not attempt them ourselves) that one probably ends up back in the flowing cone scenario. However, there may be some important insights into the ER=EPR hypothesis that can be obtained from the fact that this condensate at a codimension-2 surface is apparently equivalent to ordinary flat space.

## 6.2 Higher genus corrections

Next we discuss the prospects for extending S&U's result to new settings. Unfortunately, it is somewhat difficult to find situations in string theory where (i) we have full control over the worldsheet theory, and (ii) there is a finite sized correction to $A/4G_N$, that is neither zero nor divergent.

The first obvious correction to consider is the effects of the higher genus corrections, starting with the torus g = 1 contribution. Since the torus correction is analogous to the 1-loop correction in field theory, one expects to obtain from it a quantum l-loop correction to the black hole entropy. From a semiclassical perspective, the 1-loop correction would contribute to the $S_{\text{out}}$ term in the generalized entropy (45), and if one integrates out the leading order area term in $S_{\text{out}}$, one would obtain an additive renormalization shift of the inverse Newton's constant $1/G_N$.[46]

Unfortunately, this effect cannot be easily seen in either bosonic or superstring theory (for reasons mentioned briefly at the end of section 3). In the bosonic case, the IR problems associated with the tachyon cause the torus diagram to diverge, so one gets $\infty$ for the torus diagram. On the other hand, for superstrings there is a target space nonrenormalization theorem in $D = 10$ Minkowski which causes all higher genus diagrams with $n \leq 3$ on-shell insertions to vanish. Since $G_N$ can be measured from the graviton 3-point function, this means that it is unrenormalized and so $S_{\text{torus}} = 0$.

This is a little strange because one might have expected that the torus contribution to the von Neumann entropy $S_{\text{out}}$ is an inherently positive quantity. But negative contact terms can appear in the black hole entropy under certain circumstances. For example, in the particle ($\alpha' \to 0$) limit of string theory, a negatively contributing "contact term" in the black hole entropy was found by [79] for a $U(1)$ Maxwell field. This was later resolved in [80,81] where it realized that this term is fully explained by the entanglement entropy of *edge modes*, which can be negative in certain continuum regulator schemes.[47] Similar contact terms presumably appear for higher spin fields [62,83], although there are additional subtleties in this case (cf. [84] and references therein). It would be interesting to try to understand this cancellation from a worldsheet perspective. (In particular, it is interesting that the torus nonrenormalization theorems seem to be valid only when including edge modes and bulk entanglement terms together.)

## 6.3 Other backgrounds

The other obvious direction to modify the S&U calculation is by going to other backgrounds besides Rindler.

---

[46]See e.g. section 3.12 in [78].

[47]A similar contact term which appears for the non-minimal scalar should instead be thought of as a contribution to a Wald entropy term $\langle \phi^2 \rangle$ on the horizon [82], see [41] for an example of how such terms can arise from models where the microscopic interpretation is still an entanglement entropy.

The most straightforward extension is to consider the effects of $\alpha'$ corrections, which in general produce higher curvature corrections in the effective action $I_0$. This could be done along the same lines as section 3, but taking into account the effects of higher loop diagrams.

That being said, the effects of higher curvature entropy on the black hole entropy have already been explored extensively. It is not totally clear what is gained thinking of such calculations from a worldsheet perspective, once we know from S&U that it works at leading order.

A more interesting result would be to calculate black hole entropy in a *highly stringy regime* that is nonperturbative in $\alpha'$ where one has no choice but to think of entropy from a worldsheet perspective. It is, however, difficult to find a regime which would enable a nontrivial result. For one thing, the worldsheet theory would probably need to be understood as an exact CFT, which limits one to a very restricted class of backgrounds (in superstrings, all of them are NS-NS).

One possibility is the two-dimensional black hole [85, 86] whose Lorentzian worldsheet CFT is the group coset $SL(2, \mathbb{R})/U(1)$. The Euclidean version is the *cigar* background with a coset CFT given by $H_3/U(1)$. The cigar has an interesting set of dualities; by the FZZ correspondence, the cigar is dual to 2d ($c = 1$) sine-Liouville string [87, 88], which itself is dual to a one-dimensional matrix quantum mechanics with a single matrix [89, 90].

One of us (A.A.) was involved in a collaboration that identified the boundary microstates of this two-dimensional black hole in the *dual* matrix quantum mechanics and reproduced one of the two expressions for the free energy found in [90], at leading order in large $N$ [91]. A string theory interpretation and count of these microstates on the bulk side, specifically on the cigar, would be a natural application of the off-shell formulation of string theory presented in this paper.

## 6.4 Holographic entropy formula

Another interesting possibility is to consider a string background in a holographic AdS background. In this case, a S&U type calculation can be performed on the *bulk* side of the AdS/CFT duality, to obtain a worldsheet derivation of the holographic entanglement entropy [92, 93].

The simplest non-trivial example to consider is the pure NS-NS flux $AdS_3 \times S_3 \times \mathbb{T}^4$, which is an exact string background with a worldsheet description in the bulk. It is equivalent to an $SL(2,\mathbb{R})$ WZW model, times the compact directions. This background has been studied and analyzed extensively in the literature [94–102] shortly after the AdS/CFT duality was proposed, with a plethora of recent amazing work on the tensionless limit of the string and the symmetric product orbifold [103–112]. One can also compactify a spatial direction to obtain a BTZ black hole.

If we treat the target space as a NLSM, and consider the simplest possible holographic entropy surface (which in $AdS_3$ is just a single geodesic $\gamma$) then in this case the derivation of $S = A/4G_N$ is an almost trivial extension of the Susskind-Uglum calculation in section 4.3. Since the Euclidean spacetime is $U(1)$ symmetric around $\gamma$, this simply introduces a conical singularity at the tip and one can go off-shell as before. The only new ingredient is the Kolb-Raymond potential $B_{\mu\nu}$ (which does not however contribute directly to the entropy).

Since the CFT is exactly known, it would be interesting to compute the operator of the corresponding worldsheet WZW CFT that creates a conical singularity in target spacetime (See [113, 114] for progress in this direction.) This would enable us to compute the holographic entanglement entropy *nonperturbatively* in $\alpha'$, i.e. in a very stringy regime where we cannot use bulk field theory, including cases where the dual CFT is weakly coupled. Unfortunately this is not quite as exciting as it sounds, because in this case $S$ is proportional to the boundary central charge $c$, which is independent of $\alpha'$ by virtue of the *boundary* c-theorem. (To avoid this, one would need to find a stringy AdS which is not continuously connected to

an AdS background with small $\alpha'$, but then it is presumably difficult to have control over the worldsheet theory.)

A closely related approach is to orbifold the AdS$_3 \times$ S$^3 \times \mathbb{T}^4$ background in such a way as to break supersymmetry and thus have twisted tachyons localized at the tip of the orbifold fixed point (the cone) [115, 116]—just as we discussed for Rindler in section 5.2. This can be done by orbifolding only AdS$_3$, i.e. AdS$_3/\mathbb{Z}_N$. Here, tachyon condensation plays a major role. This orbifold approach is similar to the one considered in [39] for $\mathbb{C}/\mathbb{Z}_N$. In fact, the condensation of these tip-localized closed string tachyons in AdS$_3/\mathbb{Z}_N$ was studied numerically in [117] where they also was found that AdS$_3/\mathbb{Z}_N$ decays, by emitting a dilaton pulse that propagates to the boundary, into to AdS$_3/\mathbb{Z}_K$ with $K < N$ until it reaches the pure AdS$_3$ vacuum.

Since the holographic entropy surface $\gamma$ considered above has a U(1) symmetry, so far this derivation is akin to the Casini-Huerta-Myers derivation of stationary holographic entropy [118].

It would be very interesting however to try to extend the stringy calculation to the non-U(1) symmetric case. In that case, to calculate the *boundary* von Neumann entropy, one has to do a replica trick calculation of the boundary CFT:

$$S = (1 - \mathcal{N}\partial_{\mathcal{N}})Z[\mathcal{N}]\Big|_{\mathcal{N}=1}. \tag{80}$$

By a clever argument of Lewkowycz-Maldacena [119], on the *bulk* side of the duality, it is still possible to perform this analytic continuation in a geometrical way using an orbifold of the replicated background. (See [120, 121] for the extension of this argument to the 1-loop quantum corrections to $S_{\text{gen}}$, and [122–124] for further extensions.)

It is natural to wonder whether these arguments can be extended to the case of worldsheet string theory, perhaps using actual orbifolds. In that case, tachyon condensation at the tip may play a significant role in proving the equivalence of the orbifold background with the original replicated saddle.

## Acknowledgments

We are grateful for conversations with Edward Witten, Arkady Tseytlin, Gabriel Wong, William Donnelly, Ronak Soni, Juan Maldacena, Donald Marolf, Raghu Mahajan, Lorenz Eberhardt, Eva Silverstein, Daniel Jafferis, Xi Yin, Lenny Susskind, Alexander Frenkel, Vasudev Shyam, Ayshalynne Abdel-Aziz, Zihan Yan, Houwen Wu, David Tong, and David Skinner. A.A. would like in particular to thank Prahar Mitra for extensive, very long and insightful discussions. A.W. would also like to thank Joe Polchinski for pointing him in the direction of Tseytlin's work, several years before he had the capacity to actually understand it.

**Funding information** This work was supported in part by AFOSR grant FA9550-19-1-0260 "Tensor Networks and Holographic Spacetime", STFC grant ST/P000681/1 "Particles, Fields and Extended Objects", and an Isaac Newton Trust Early Career grant.

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
