# Peer review of "Off-Shell Strings II: Black Hole Entropy"

_SciPost Physics, doi:SciPost Phys. 17, 006 (2024)_

## Round 1 · Referee Report · Lorenz Eberhardt (Referee 1) · 2023-9-27

Report

The authors make a brave attempt at one of string theory's most notorious problems: to give a general direct derivation of Hawking's black hole entropy formula $S_\text{BH}=\frac{A}{4G}$. The paper does not contain major new ideas, but instead clarifies several existing approaches and puts them into perspective, as well as discusses their strength and weaknesses. The literature on this subject is filled with vague and partially contradictory statements and the authors do a great job of informing the reader about their validity from a modern point of view. The main approach that the authors pursue is a stringy uplift of the derivations of black hole entropy from the gravitational path integral. This entails computing the leading contribution of order $\frac{1}{G}$ to the string partition function, which is captured by the sphere in perturbative string theory. Computing the sphere partition function of a string worldsheet theory is a subtle problem and the authors use the technology developed in their previous paper which in turn is based on old papers of Tseytlin. In those papers, it is proposed to allow off-shell string backgrounds, i.e. string backgrounds that do not satisfy the equations of motion and which are not represented by CFTs on the worldsheet. This requires the introduction of a UV-cutoff $\varepsilon$ on the worldsheet and the effect of dividing by the volume of the Moebius group is roughly realized in the off-shell approach by taking a derivative with respect to $\log \varepsilon$. The authors use this prescription and demonstrate that one can derive the full off-shell spacetime effective action order by order in $\alpha'$ from the formalism. They carry this program explicitly out to first order in $\alpha'$ (which already involves 2-loop computations in the dilaton sector).

I greatly appreciate that the authors attack such a difficult problem in string theory. Even though I would argue that their main contribution is to flesh out and compare various partially existing proposals and computations in the literature, I think that this paper is a very valuable resource for the community. It is a bit anti-climactic that the computation is essentially reduced from a string theory computation to a gravity computation and is thus not `inherently stringy'. In particular these methods will presumably not give new insight into the nature of black hole microstates in quantum gravity. I also thought that although one of the main goals of the paper was to clarify Susskind's \& Uglum's open string picture, they did not add much more to it. I think that this paper is suitable for publication in SciPost.

I have some mostly minor remarks:

  1. Eq. (8): I don't understand the last step, why is it still $z$-dependent? Is this some sort of zero mode of $\log G(X(z))$?
  2. The first term in eq. (10) should presumably read $\frac{1}{4\pi \epsilon^2}$.
  3. Page 9, last paragraph of section III: Please provide a reference for the mentioned renormalization theorem (the same appears again in the discussion section VI.B without reference).
  4. Please also provide a reference for eq. (45). In what sense are the expectation value brackets around $\frac{A}{4G}$ to be interpreted?
  5. Eq. (53): $t \to \tau$
  6. Paragraph after eq. (60): The claim that RG flow on the worldsheet corresponds to a Ricci flow in the target manifold should be explained better. I would have expected a generalized version of a Ricci flow, since there is also a dilaton field that flows (the authors put the B-field to zero by assumption). It should also be mentioned that this statement is only true to leading order in $\alpha'$, there would be corrections at higher orders. It should perhaps also be explained that this statement is true in the string frame, not the Einstein frame. This distinction is never made in the article.
  7. Eq. (63): Please think about using another variable than $\beta$, as it can be very easy to mix up with the inverse temperature.
  8. Eq. (65): I think the biggest issue with such a factorization is that once we take back reaction into account, the inner and outer regions cannot be defined anymore since the horizon itself fluctuates. Thus (65) can at best only hold approximately in the limit of weak string coupling.
  9. The explanation before eq. (68) is in my view incorrect or misleading. The embedding of the worldsheet into Euclidean target space $X: \Sigma \longrightarrow \mathfrak{M}$ is always a continuous map (but not necessarily differentiable). The horizon $H \subset \mathfrak{M}$ is a closed subset in the Euclidean spacetime. Thus by simple topology $X^{-1}(H)$ is closed and hence compact (since $\Sigma$ is compact). Since $\Sigma$ intersects $H$ generically in a number of points, $X^{-1}(H)$ consists of a number of points and is thus by compactness finite. Thus the worldsheet intersects the horizon actually only a finite number of times. Consequently, I also think that the following discussion about adding stiffness terms is moot. (Incidentally the inconsistent use of $m$ and $n$ around eqs. (69) and (70) is also confusing to the reader.)
  10. Section V.A: The argument that the sphere contribution in the on-shell approach to the orbifold replica trick vanishes is unconvincing to me. The authors point to their previous paper, section II.B for this. The argument their is a supergravity argument which shows that for smooth and compact target spaces, the sphere partition function vanishes since the action is on-shell a total derivative. This argument does not apply to the orbifold, since the orbifold is neither smooth nor compact. As far as I am aware it is not known how to compute the sphere contribution and it is currently unknown whether it is zero or non-zero. The only reliable argument for a vanishing sphere partition function that I am aware of only works at $N=1$, where at least for the superstring, spacetime supersymmetry requires a vanishing on-shell action. I urge the authors to be more honest about the comparison. It is fair to say that the on-shell orbifold approach is not able to produce quantitative results for the sphere contribution at the moment.
  11. The first sentence of the last paragraph on the left column of page 17 presumably has a typo and I don't understand what the authors want to say.

Requested changes

Please address my remarks.

  • validity: high
  • significance: top
  • originality: good
  • clarity: high
  • formatting: perfect
  • grammar: perfect

Author:  Amr Ahmadain  on 2024-05-20  [id 4497]

(in reply to Report 1 by Lorenz Eberhardt on 2023-09-27)
Category:
question
answer to question
reply to objection

Dear Lorenz,

We have addressed all of your comments, remarks and questions.

Please see attached PDF file for a detailed exposition of all changes made to the text. Our replies are in blue.

If you still have any further questions or comments, we will be happy to address them.

The Authors

Attachment:

Off_shell_Strings_II_SciPost_Response_to_Lorenz_Eberhardt.pdf

---

## Round 1 · Referee Report · Indranil Halder (Referee 2) · 2024-2-19

Report

Understanding the closed string origin of thermal entropy (and time-dependent generalizations thereof) of a black hole is arguably the most prestigious and challenging open question of string theory in the twenty-first century. Authors of the draft have attempted to review the existing literature on the subject in a coherent, modern language. The central topic of the article is the discussion of the Susskind and Uglum type off-shell method for the evaluation of Bekenstein and Hawking entropy, whose most precise version is the replica trick of Lewkowycz and Maldacena.
1. The method is based on target space physics and inherently perturbative in $\alpha'$. The authors concentrated on the leading order in the $\alpha'\to 0$ limit. In sections II, III the authors presented clear calculational details of the target space effective action from the $\beta$ function of the non-linear sigma model type worldsheet based on the old work of Osborn and Tseytlin. 2. The central value of the draft lies in the sections IV, B, and V, A. In section IV, B authors strongly emphasized that the off-shell method of the replica trick gets the contribution to blackhole entropy from the bulk Einstein-Hilbert type term only. While this was already expected in the paper by Susskind and Uglum itself and the argument presented in the draft is not complete, the statement is indeed correct and crucial for understanding the boundary effect in stringy theory as explained in the recent work of Halder and Jafferis. In section V, the authors emphasize that if the back-reaction of the twisted sector tachyons is not considered in the orbifold method then the contribution to Bekenstein-Hawking entropy vanishes. This generates a very important question: what are we calculating in the orbifold method of Dabalkar and Witten at torus or higher genus worldsheets? A plausible suggestion for the answer appeared in the later works of Halder and Jafferis.

While there is a lot to improve on the level of detail and rigor of the the draft, in my opinion, the article presents modern emphasize on a very important question of high energy physics and qualifies to be published in SciPost.

  • validity: -
  • significance: good
  • originality: low
  • clarity: low
  • formatting: good
  • grammar: good

Author:  Amr Ahmadain  on 2024-05-20  [id 4498]

(in reply to Report 2 by Indranil Halder on 2024-02-19)
Category:
remark

Dear Indranil,

We thank you for your comments. We have cited your recent work with Daniel Jeffries on the stringy replica trick in the appropriate places within the text.

The Authors

---

## Round 2 · Author Response

Dear Editor-in-charge,

Here is the revised version of the manuscript which contains all the changes described in the PDF files attached as individual replies to the referees.

We kindly ask you to recommend the paper for publication to the editorial college.

Best regards,
The Authors

---

## Round 2 · List of Changes

Please see the PDF files for detailed exposition of the changes made to the text. The PDF files should be publicly visible.

---

## Editorial Decision

published